# Noncovalent Complexes of Cyclodextrin with Small Organic Molecules: Applications and Insights into Host–Guest Interactions in the Gas Phase and Condensed Phase

**DOI:** 10.3390/molecules25184048

**Published:** 2020-09-04

**Authors:** Jae-ung Lee, Sung-Sik Lee, Sungyul Lee, Han Bin Oh

**Affiliations:** 1Department of Chemistry, Sogang University, Seoul 04107, Korea; lju2529@gmail.com; 2Department of Applied Chemistry, Kyung Hee University, Gyeonggi 17104, Korea; int_shik@hotmail.com

**Keywords:** cyclodextrin, host–guest chemistry, mass spectrometry, ion mobility spectroscopy, IRMPD spectroscopy, DFT calculation

## Abstract

Cyclodextrins (CDs) have drawn a lot of attention from the scientific communities as a model system for host–guest chemistry and also due to its variety of applications in the pharmaceutical, cosmetic, food, textile, separation science, and essential oil industries. The formation of the inclusion complexes enables these applications in the condensed phases, which have been confirmed by nuclear magnetic resonance (NMR) spectroscopy, X-ray crystallography, and other methodologies. The advent of soft ionization techniques that can transfer the solution-phase noncovalent complexes to the gas phase has allowed for extensive examination of these complexes and provides valuable insight into the principles governing the formation of gaseous noncovalent complexes. As for the CDs’ host–guest chemistry in the gas phase, there has been a controversial issue as to whether noncovalent complexes are inclusion conformers reflecting the solution-phase structure of the complex or not. In this review, the basic principles governing CD’s host–guest complex formation will be described. Applications and structures of CDs in the condensed phases will also be presented. More importantly, the experimental and theoretical evidence supporting the two opposing views for the CD–guest structures in the gas phase will be intensively reviewed. These include data obtained via mass spectrometry, ion mobility measurements, infrared multiphoton dissociation (IRMPD) spectroscopy, and density functional theory (DFT) calculations.

## 1. Introduction

Cyclodextrins (CDs) are macrocyclic oligosaccharides commonly composed of six, seven, or eight d-glucopyranoside units linked in α-(1,4) bonds (Figure 1), denoted as α-CD, β-CD, and γ-CD, respectively. The main characteristics of α-, β-, and γ-CDs are given in Table 1 [1,2,3]. Although these compounds were discovered in 1891 by Villiers et al., an accurate representation of their structure was only determined 50 years later via X-ray crystallography [4]. The unique structure and its functionality to form inclusion complexes have become the focal point of scientific and commercial efforts [1,2,3]. CDs generally possess a truncated cone with a cavity whose hydrophobicity is due to the presence of the C3 and C5 hydrogens and a glycosidic oxygen atom. The external face of these compounds is hydrophilic. At the wider rim (i.e., the secondary rim), there are hydroxyl groups on the C2 and C3, whereas the narrower rim (i.e., the primary rim) possesses a C6 hydroxyl group [5,6,7]. These structural characteristics facilitate the use of CDs in a wide range of chemical and analytical applications, namely, in sample preparation [8,9,10], purification [11,12,13,14,15], spectroscopic analysis [16,17], as booster agents for enhancing analyte sensitivity during luminescence and fluorescence spectroscopy, templates for artificial enzymes [18,19], and biosensors [20,21,22,23]. The most notable feature of these compounds is their ability to selectively form noncovalent host–guest complexes with small molecules or enantiomeric species [1,2,3] due to molecular and chiral recognition processes, respectively. Here, the preferential interaction of a CD with a specific molecule or enantiomer serves as the backbone for reactions related to size and structure complementarity. The driving forces behind the interaction between CD and the individual guest molecules include electrostatic, van der Waals, hydrogen bonding, and hydrophobic interactions, as well as the quest for relief from conformational strain and high energy [24,25]. Molecules in the gas phase undergo predominantly polar interactions, such as hydrogen bonding and electrostatic interactions [26], which is due to the absence of the solvent shielding by water molecules. On the other hand, the hydrophobic interactions are weakened in the gas phase, in which the absence of water removes the driving force for the gathering of nonpolar surfaces [27,28]. These changes in interactive forces imply that the direct comparison of noncovalent interactions (or complexes) between the gas and condensed phases should be cautious, in particular for the quantitative determination [27,28,29].

The so-called “three-point rule” or “lock and key model” has been applied to elucidate the mechanism governing molecular/chiral recognition of CDs in the solution phase [30]. In the three-point rule applied to the CD complex system, the three points on the complex are composed of one hydrophobic point and two interaction points (including hydrogen bonding, Coulomb interaction, and van der Waals repulsion) [24,31]. For example, the complexation of CD with aromatic amino acids can be explained by the three-point rule, where aromatic amino acids have one hydrophobic point in the side chain and two hydrogen bonding points of CO^2−^ and NH^3+^ groups. The lock and key model, which was first suggested by Emil Fisher, attributes the specific binding of enzyme and substrate to the optimal geometric fit between them [32]. This model is especially useful for explaining the recognition of guest molecules lacking notable interaction parts in the CD complex system [24]. Various strategies have been utilized to elucidate the interaction mechanism, namely, nuclear magnetic resonance (NMR) spectroscopic techniques (e.g., nuclear Overhauser effect (NOE) and rotating-frame Overhauser enhancement (ROESY) spectroscopy) and theoretical calculations (e.g., molecular dynamics (MD), semi-empirical, and density functional theory (DFT) methods) [33,34,35,36].

In attempts to elucidate the specific mechanism governing CD recognition of small molecules in the gas phase, researchers have noted that the three-point interaction or lock and key model can be applied putatively, as used in the case of the solution phase, even though there is no exact model to explain all different cases of CD–guest complexes [37]. These complexities can also be found in the noncovalent interaction of CD/inorganic guest molecules, including metal cations [38,39], polyoxometalates [40,41,42,43], dodecaborates [44,45,46], and octahedral rhenium clusters [47,48]. The three-point rules are not applicable to explain the complexation of these molecules, in which more than three interaction points could exist or no hydrophobic interaction takes place. Other references suggest that the complementarity of the cavity size is also an important factor for the complexation. Mechanism studies for CD complexes in the gas phase have been performed using a combination of the mass spectrometric methods, namely, gas chromatography–mass spectrometry (GC–MS), ion mobility mass spectrometry (IMS–MS), infrared multiphoton dissociation (IRMPD) spectroscopy, and theoretical calculations involving various combinations of molecular dynamics (MD), semi-empirical, and density functional theory (DFT) methods [2,26,49,50,51,52,53,54,55,56,57,58,59,60,61]. Recently, structural analysis was conducted using action–FRET (Fluorescence Resonance Energy Transfer), a gas phase version of FRET in which chromophore-tagged β-cyclodextrin and amyloid-β fragment peptides were utilized [61].

Herein, a review of the applications (in the solid/solution phases), structural analysis, and the noncovalent interactions of CDs in the gas phase are presented, focusing on the gas phase noncovalent interactions between CD and the guest molecules. Thus far, two opposing schools of thought govern the definitions and characteristics of gas-phase CD–guest interactions. One theory states that a hydrophobic compound is readily incorporated into the hydrophobic cavity of the CD. Conversely, the other theory proposes that the relevant interactions are predominantly due to hydrogen bonding between the hydrophilic functional groups along the outer rims of the CD, with little to no contribution from the interior hydrophobic cavity. In this review, experimental and theoretical evidence in support of these differing viewpoints will be presented using data gathered via mass spectrometry, ion-mobility measurements, IR spectroscopy, and density functional theory (DFT) calculations (see below).

## 2. Applications of CD Inclusion Complexes in the Solid/Solution Phases

CDs are particularly useful agents for pharmaceutical, cosmetic, food, textile, separation science, and essential oil industries due to their biocompatibility, chirality, and unique ability to host a range of guest molecules in hydrophobic cavities by forming inclusion complexes. Encapsulating a guest molecule enables better control of the characteristics of the guest molecule, including its solubility, volatility, permeability, and chemical reactivity. In this section, we have opted to represent state-of-the-art industrial applications of CDs to emphasize the importance of CDs for the above-mentioned fields, in particular in the solid/solution phases, implying the significance of precise characterization of CD complex structures in the gas phase too.

### 2.1. Pharmaceuticals

CDs are commonly found in more than 84 pharmaceutical products for oral, nasal, rectal, dermal, ocular, and parenteral formulations [3,62,63]. Derivatized CDs improve drug bioavailability as a pharmaceutical excipient or drug delivery vehicle by increasing solubility, stability, permeability, and absorption of the therapeutic agent while reducing toxicity [63,64,65]. Furthermore, CDs function as active pharmaceutical ingredients (APIs) in some cases. This is best exemplified by Sugammadex (also known as Bridion^®^), which is a modified γ-CD (octakis-(6-deoxy-6-*S*-mercaptoproprionyl CD sodium salt) used as a reversal agent for curare-like compounds such as rocuronium and vecuronium in anesthesia [3,66,67]. This drug’s effect is derived from the selective encapsulation of the curare-like agents, which causes the steroidal neuromuscular blocker to lose its ability to bind to the acetylcholine receptor, thereby facilitating rapid excretion via the kidneys [3,66]. This trait makes CDs ideal cholesterol regulators and therapeutic agents for various diseases, including atherosclerosis, cancer, Niemann-Pick Type C, and Alzheimer’s [3,68]. One example is the therapeutic agent Trappsol^®^ Cyclo™ (Cyclodextrins CTD), which is currently undergoing Phase III trials to determine its safety and efficacy for the treatment of Niemann-Pick Type C disease [69].

### 2.2. Cosmetics

Their ability to encapsulate guest molecules makes CDs superb agents for commercial cosmetic applications. This trait endows a plethora of advantages to the CD-encapsulated cosmetic product, including improved physical and chemical stability against light and oxidation, more controlled release of volatile compounds and fragrances, increased solubilization of insoluble components, easier removal of unpleasant odors or undesired external smells, a straightforward modification process for the formulation from liquids or oil powder, improved skin penetration capabilities, and slowed release of the active ingredients via the application of nanotechnology or nanosystems [70,71,72,73,74]. Recently, Mori et al. reported that the complexation of HP-γ-CD/trans-ferulic acid (t-FA) improved the anti-UV properties of eco-friendly sunscreens and increased the solubility of t-FA, an all-natural UV-absorbing ingredient derived from rice bran [75]. CDs are of particular interest to manufacturers in emerging applications for nanotechnology-based and eco-friendly cosmetic products [74,75].

### 2.3. Foods

The incorporation of CDs in the food manufacturing industry has allowed manufacturers to accomplish previously unfathomable tasks, such as protecting volatile aromatics during food processing or storage, eliminating or masking unpleasant odors or tastes, and stabilizing colors, tastes, and flavors by protecting the food products against heat, humidity, and oxidative reactions. CDs also serve as effective food packaging materials in antimicrobial delivery systems that enable long-term storage and preservation [76,77,78,79,80,81,82,83]. CDs are of nutritional benefit as they facilitate the delivery of extremely insoluble nutraceuticals, such as omega-6 and omega-3 fatty acids, phytosterols, and vitamins [84], help improve the absorption of nutraceuticals, remove cholesterol from milk and egg products to obtain “cholesterol-reduced” foods [85,86,87], and decrease the postprandial blood levels of sugars, lipids, and cholesterol for health and body weight management purposes [88,89,90]. The global trend towards a healthier diet and lifestyle has created countless opportunities for the use of CD-based products as dietary supplements [84].

### 2.4. Textiles

Over the last few decades, clothing manufacturers have been employing CDs in a variety of textile manufacturing processes to impart new functionalities to natural textiles, improve the quality of the textile, and facilitate eco-friendly production by replacing extremely polluting processes that are typically associated with conventional textile manufacturing, such as dyeing, printing, and textile finishing [91]. The covalent conjugation of CDs into the fabric’s framework enables producers to strategically incorporate useful properties, such as odor-fighting traits, antimicrobial resistance, antibiotic action, and UV protection, by using the appropriate guest molecule [72,92,93]. The use of CDs during the textile dyeing process offers the advantage of retarding “overdyeing” (i.e., the uneven adsorption of excessive amounts of dyes) since the incorporated CDs directly compete to capture the dye molecules, thereby ensuring the slow absorption of dye molecules and more uniform coloring throughout the textile [94]. Furthermore, CDs are integral in the washing/rinsing stages of the finishing process as they are used to eliminate unpleasant odors or harmful residues from the dyeing process. Lastly, CDs are vital for the end-stage purification of the wastewater produced during the textile manufacturing process [95]. Since the use of CDs during the textile manufacturing process offers an environmentally friendly approach for the production of functionalized fabrics, exponential growth in this field is anticipated in the future.

### 2.5. Separation Science

CD-containing chiral stationary phase (CSP) is extensively used in gas chromatography (GC), high-performance liquid chromatography (HPLC), thin-layer chromatography (TLC), micro solid phase extraction (μ-SPE), capillary electrophoresis (CE), capillary electrochromatography (CEC), electrokinetic chromatography (EKC), and supercritical fluid chromatography (SFC) for the separation of structurally similar compounds or complex product matrices of enantiomers [96,97,98,99,100,101]. The derivatization or coupling of CDs using various functional molecules is also widely employed in this field [102,103,104,105,106,107,108,109,110,111,112]. One such example can be seen in the study conducted by Sun et al., in which per-4-chlorophenylcarbamate/β-cyclodextrin-based CSP was developed and successfully applied for the separation of flurbiprofen enantiomers and 22 other drug racemates [109,110]. Xu et al. prepared packed columns containing γ-CD/metal–organic framework (MOF) complexes that exhibited efficient drug-loading capabilities [103].

With the advent of ultra-high-performance liquid chromatography (UHPLC), the use of sub-2 Micron CSPs with integrated CDs became increasingly popular in the past decade [97,113,114]. In 2017, Silva et al. applied 3,5-dimethylphenylcarbamoylated β-CD to amino-functionalized silica in nano/LC and CEC systems for the effective enantioseparation of flavonoids and flavones [115]. Recently, numerous papers were published on the application of cyclodextrin-modified electrokinetic chromatography for rapid enantioseparation of uncharged or partially charged molecules [116,117,118]. The developments detailed in the above-mentioned studies indicate that the incorporation of novel CD-based technologies for enantioseparation applications is a promising field for future researchers.

### 2.6. Essential Oils

Essential oils are terpene and terpenoids-like volatile compounds presenting strong odors, which are generated from aromatic plants. These oils have many uses in the medical, food, cosmetic, and therapeutic industries because of their antifungal/antimicrobial/antioxidant functions, fine flavor, and analgesic/sedative properties. However, the oils have drawbacks such as high volatility, poor solubility, and poor stability against heat, light, or oxidation. CDs have become a useful strategy to overcome these drawbacks by enhancing physicochemical properties of these oils using inclusion complexes. A wide variety of applications using complexes composed of CD and essential oils have been recently reviewed [119,120]. Still, the extensive research of CD/essential oil complexes has been conducted to improve the properties of essential oil [121,122,123,124,125,126]. For example, Abril-Sánchez et al. presented the improved efficacy of the essential oil, citronellal, by encapsulating it using HP-β-CD [121]. As expected, the encapsulation increased solubility, and the enhanced durability was also confirmed by the GC/MS and sensory analysis. Additionally, when citronellal, HP-β-CD, and Glucobay^®^ (an antidiabetic drug) were mixed to prevent the growth of antimicrobials, the synergetic effect was observed with a long-lasting effect. The complex structure of CD nanosponge (defined as CD polymers) and cinnamon oil was also evaluated as food packaging components which have an antibacterial effect [123]. In the referred paper, the CD nanosponge/cinnamon oil complex showed a better antibacterial effect than cinnamon oil alone, even at a lower amount with the increased stability.

## 3. Structures of CDs and Their Derivatives in the Solution- and Solid-Phases

### 3.1. Symmetrical CDs

Since numerous CD conformers are equally feasible both in the gaseous and solution phases due to similarities in their Gibbs free energies, the unambiguous assignment of the respective structures is quite challenging. A most useful guide would be X-ray-determined structures, but no agreement with the solution-phase CD structures could be guaranteed. Quantum chemical calculations would also be a good help. Wolschann and co-workers carried out quantum chemical calculations for gas-phase symmetric α-, β-, and γ-CDs by the B3LYP/6-31G(d,p) method [127]. The minimum energy conformers (Figure 2) and the associated energy profiles were determined by varying the distances between the oxygen atoms of the primary hydroxyl groups. Global minimum energy conformers such as α, β, and γA, which are commonly referred to as the A series, exhibited small O6–O6′ distances of 2.77, 2.80, and 2.85 Å for α-, β-, and γ-CDs, respectively. Here, the two hydrogen-bonded rings adopted basket-like conformations in which one intramolecular hydrogen-bonded ring included the primary hydroxyl groups, and the other encompassed the secondary hydroxyl groups of the incorporated CDs. The local minimum energy conformers of α, β, and γC exhibited much larger O6–O6′ distances, namely, 6.38, 6.11, and 5.96 Å for α-, β-, and γ-CDs, respectively. These conformers possess significantly higher energies than the previously mentioned A-series as a consequence of the missing hydrogen bonding interactions at the primary hydroxyl groups. The authors of that study determined that the C conformers were in better agreement with the crystallographic geometries, even though the latter structures were asymmetric and exhibited quite significant deviations in the oxygen–oxygen bond distances due to the influence of nearby water molecules.

Jáuregui-Haza et al. employed the SMD/M06-2X/6-31G(d,p) DFT method to elucidate the structures of α-, β-, and γ-CDs in solution (Figure 3) [128]. Here, the conformers were subsequently classified into three groups, namely, A, B, and C. As with Wolschann’s group, Jáuregui-Haza et al. focused on distinguishing the conformers based on the distances between the –OH groups of the hydrogen bonding interactions. The A-type conformers were characterized by O6–H···O6′ hydrogen bonding interactions with the primary hydroxyls oriented toward the cavity, thereby forming a ring of hydrogen bonds between adjacent primary hydroxyls. The B-type conformers had O6–H···O5′ hydrogen interactions, whereas the C-type conformers possessed primary hydroxyls pointing to the exterior of the CD that exhibit hydrogen bonding interactions with the O5 of the same glucopyranose unit (i.e., O6–H···O5). The researchers also compared the calculated structures obtained with those determined via X-ray diffraction experiments, and found that the values between the two were in good agreement with each other; here, the root means square deviations of the bond lengths and angles were always less than 2% [129,130,131].

Several studies were devoted to determining the crystal structures of CDs containing water molecules [131,132,133,134,135]. In 1974, Manor and Saenger used X-ray diffraction to determine the structure of α-CD compounds containing six water molecules. The researchers noted that four of the six water molecules were located outside the CD. In contrast, the remaining two water molecules could be found inside the aperture of the CD molecule along the molecular axis [132]. These two water molecules were hydrogen-bonded to each other, i.e., the water molecule located closer to the O(6) side of the macrocyclic ring was hydrogen-bonded to two O(6) hydroxyl groups (Figure 4).

Saenger et al. studied the structure of β-CD·(H_2_O)_11_ via neutron diffraction [136] and determined that eight of the eleven water molecules distributed across sixteen positions were located inside the β-CD cavity (i.e., 6.13 water molecules) and eight were found in the interstices (i.e., 4.88 water molecules). The cavity-bound water molecules exhibited two hydrogen bonding interactions with the β-CD molecule and six contact points with the neighboring β-CD molecule (Figure 5). The experimental scope of the X-ray and neutron diffraction studies were subsequently extended to facilitate the structural elucidation of β-CD molecules containing both guest and water molecules, as noted in compounds like the partially deuterated β-CD ethanol octahydrate. This allowed the researchers to accurately document the extensive hydrogen bonding networks surrounding the β-CD molecule [137]. Steiner and Koellner conducted structural elucidation studies to determine the effect of humidity on the structure of the crystalline β-CD hydrate [133].

### 3.2. Permethylated β-CDs

Derivatized CDs such as permethylated CDs (per-CDs), in which all –OH moieties had been substituted by –OCH_3_, are useful for elucidating the nature of the host–guest interactions. When the guest molecule contains hydroxyl groups, e.g., amino acids, their IR absorption bands tend to interfere with the CD absorption bands. Permethylation of the CD molecules simplifies the analysis and enables easier assignment of the IRMPD spectra. Previously, Oh, Lee, and colleagues determined the structures of per-β-CD by conducting canonical ensemble (NVT) molecular dynamics simulations based on the four initial backbone structures of β-CDs containing hydroxyl groups that had been computationally predicted by Wolschann et al. [56,127]. Here, the two lowest Gibbs energy structures, as depicted in Figure 6, of the per-β-CD molecules were employed for the structural elucidation of the per-β-CD/H^+^/H_2_O complex whose IRMPD spectra had been obtained by Oh et al. [56]. A comparison of the calculated IR spectra with the experimentally observed IRMPD spectra, as well as the processes behind the structural determination of the gaseous per-β-CD/H^+^/H_2_O system and the per-β-CD/H^+^/amino acid complexes, will be discussed in subsequent sections.

## 4. Noncovalent CD Complexes in the Gas Phase

### 4.1. Mass Spectrometry

Since it was first shown that electrospray ionization mass spectrometry (ESI-MS) could be used to detect noncovalent complexes by Ganem et al. [138,139], a large number of publications have been reported studying the noncovalent CD complexes by ESI-MS [140,141,142,143]. To date, various studies were made for the analysis of noncovalent CD complexes, including amino acids, peptides, drugs, steroid hormones, and simple hydrophobic compounds [140,141,143,144,145,146,147]. Even though mass spectrometry has become the most convenient and rapid analytical platform for the investigation of non-covalent complexes with high sensitivity and selectivity, there have been some controversial results about the structure of CD/guest noncovalent complexes in the gas phase. Initially, CD-small-molecule noncovalent complexes were regarded as inclusion complexes that contained a small molecule within the cavity [148,149]. Thanks in part to NMR analysis of the associated solutions, the idea of the inclusion CD complexes in ESI-MS studies was widely accepted without any rigorous scrutiny, even though the ionization/transfer process can change complex structures [150,151,152], until Cunniff and Vouros highlighted the possibility of false-positive detection in the ESI-MS studies [141]. In their study, both aromatic amino acids (i.e., tyrosine, phenylalanine, and tryptophan) and nonaromatic basic amino acids (i.e., lysine, arginine, and histidine) readily formed noncovalent complexes with β-CD in ESI–MS analysis. These findings raised suspicion as it was highly unlikely that basic amino acids would be included within the cavity of β-CD in solution [6], thereby proving that false-positive-inclusion CD complexes could be observed during ESI-MS analysis. In light of this, control studies were conducted to confirm the formation of an inclusion complex and to determine if an electrostatic adduct artifact had been generated due to the ESI process. Gabelica et al. also suggested the formation of nonspecific adducts of α-CD/linear α,ω-dicarboxylic acids (diacid) in the gas phase [153]. It is well known that the association constant of this complex in the solution phase is linearly correlated with the chain length (hydrophobicity) of the diacid, which verifies the formation of the inclusion complex by the hydrophobic interaction [153,154]. However, the mass spectrometry studies on the varying chain lengths showed a discrepancy from this trend: (1) in the MS spectra, there were no significant differences in the intensities of the observed ions with varying chain lengths, and (2) in the survival yield curve in tandem mass spectrometry, the shorter chain length of diacid showed better stability. They suggested that these results came from the contribution of electrostatic adducts. On the other hand, α-CD/diacid complexes showed stronger interactions compared to those between maltohexaose/diacid complexes, implying that the hydrophobic interaction exists for the formation of αα-CD/diacid complexes. Some seemingly inconsistent results were explained by the interplay of inclusion complexes and nonspecific adduct formation. By comparing the results of maltohexaose/diacid complexes with α-CD/diacid complexes, they calculated each portion of inclusion complexes and nonspecific adducts concerning the chain length (Figure 7) [155,156].

The Lebrilla group published a series of papers showing that gaseous noncovalent CD complexes adopted an inclusion complex structure [37,50,51], citing the enantioselective differentiation of a guest molecule by CD as proof of the formation of a gaseous inclusion complex. This suggestion was based on the general concept that chiral differentiation in solutions required the presence of an inclusion complex, and that the “three-point model” was better suited for the formation of an inclusion complex rather than an exclusion complex [157,158,159,160]. Based on this assumption, the enantioselectivity of CDs was investigated via the guest exchange experiment using enantiomeric amino acids complexed with CDs [50,51]. In this experiment, the complexes trapped in the analyzer were allowed to react with alkylamines that had leaked into the analyzer, and the rate constant of the guest molecule exchange process was obtained for each complex by applying pseudo-first-order rate reaction kinetics: ln *I*/*I_0_* versus t plots are used to get the slope, namely the rate constants of the complexes, where *I* is the sum of intensities of the CD/amino acid and CD/alkylamine at time t and *I_0_* is the intensity of CD/amino acid at time t_0_. Next, the enantioselectivity of CDs for chiral amino acids was determined by calculating the ratio of the rate constants of the enantiomeric complexes, i.e., *k*_L_/*k*_D_. Here, the increase in the enantioselectivity of the complex was dependent on the amino acid side-chain and followed the order: alanine (1.6) < valine (3.1) < leucine (3.6) ≤ isoleucine (3.8). Exceptions to this rule were noted for phenylalanine (0.82) and tyrosine (0.67), which were attributed to the size of the side chains. Since relatively small amino acids possessed small, flexible side-chains that could fit the CD cavity, increasing the size of the side-chains increased steric hindrance and generated “three points” of the enantiomers (i.e., the amine, carboxylic acid, and side-chain) that were an exact fit inside the cavity. Conversely, amino acids such as phenylalanine and tyrosine possessed relatively large, rigid side-chains that made positioning inside the upper rim of the cavity difficult since the carboxylic acid and amine moieties had to be forced into place in the cavity. This, in turn, resulted in reduced enantioselective interactions because the positions of the “three points” of the enantiomers were not properly aligned. This theory of size-dependent enantioselectivity of the cavity was validated via experiments involving γ-CD and partially methylated β-CD. When γ-CD was used, the researchers observed the reverse trend. In other words, the enantioselectivity was higher in phenylalanine (1.8) than in alanine (1.3) and valine (1.4). This result was consistent with the above-mentioned explanation, as the cavity size of the permethylated β-CD was large enough to accommodate alanine and valine, but was small for phenylalanine. Experiments using the partially methylated β-CD clearly indicated that cavity size was a major factor in determining enantioselectivity.

Likewise, the reflection of the solution phase to the gas phase of CDs non-covalent complexes was presented by Guo et al. [161]. They investigated the noncovalent complexes of α-, β-, and γ-CD/rutin of the gas phase using competition experiments, dilution test, and collision-induced dissociation (CID) experiments in an ion trap. The binding competition between three CDs and rutin showed different binding constants, but it was also observed that these values were corresponding to the ones of the solution phases. In dilution tests, the peak intensities of each complex did not change much following dilution, implying that the contribution of nonspecific binding is small. CID experiments also showed corresponding stability of each complex to the binding constants in the solution phase. Interestingly, the fragmentation pattern showed the selective cleavage between quercetin and rutinosyl moieties, confirming the encapsulation of quercetin moiety in the hydrophobic cavity of CDs. The fragmentation of γ-CD/rutin showed that the breakage of noncovalent interaction without internal cleavage resulted from the inefficient binding due to size difference.

The two opposite trends were noted during electron capture dissociation (ECD)-MS experiments conducted on permethylated or intact β-CD/peptide noncovalent complexes [53,162]. ECD-MS is a unique fragmentation technique that conserves labile noncovalent bonds while facilitating extensive peptide backbone fragmentation. Lee et al. proposed that the noncovalent binding of permethylated β-CD/peptide complexes was mostly due to ion–dipole hydrogen bonding rather than the hydrophobic interactions often noted in the inclusion complex of the solution-phase [53]. ECD-MS experiments on permethylated β-CD noncovalent complexes containing five different peptides revealed that the noncovalent binding observed between the permethylated β-CD and a part of each peptide had survived, while multiple fragmentations were noted within each peptide. The binding amino acid residue site between the permethylated β-CD and the peptide could be observed due to the cleavage positions in the peptide. Interestingly, the binding amino acid residue was a protonated amino acid, typically arginine. Interactions between the permethylated β-CD and the hydrophobic amino acid were not observed. Based on these findings, it was concluded that the permethylated β-CD formed an ion-dipole complex with the peptide instead of the inclusion complex, as previously believed.

Qi et al. reported that β-CD/peptide noncovalent complexes formed gaseous inclusion complexes, even though the main driving force of complexation was hydrogen bonding [162]. Similarly, the conservation of noncovalent and specific binding sites through ECD-MS analysis of the β-CD complex with six peptides was investigated. Researchers determined that amino acids, such as Arg, Tyr, Lys, Asn, Gln, and Pro, actively participated in the observed noncovalent binding interactions, and were capable of undergoing hydrogen bonding. Further evidence of hydrogen bonding interaction in these complexes came via the generation of unusual [peptide + 3H]^+^ ions after separation of the complexes, which was attributed to the abstraction of hydrogen at the hydrogen bonding position (Figure 8). From these results, the researchers concluded that the complexes bearing inclusion structures underwent hydrogen bonding interactions, thereby confirming that these complexes were not formed via electrostatic interactions. It is worth mentioning that both Lee’s and Qi’s groups obtained almost identical ECD-MS spectra but arrived at vastly different conclusions. The difference reported seemed to be a product of the difficulty associated with precisely defining the characteristics of an inclusion complex. Lee proposed that inclusion complexes encompassed a significant portion of the guest molecule within a cavity, whereas Qi’s concept of inclusion complexes was more loosely defined as complexes that exhibited hydrogen-bond-driven host–guest molecular interactions.

The disagreement about the structures of CD–guest noncovalent complex in the gas phase can be also found in the matrix-assisted laser desorption/ionization mass spectrometry (MALDI-MS) experiments. Lehmann et al. presented that the noncovalent CD–guest (PheGlyGly, TrpGlyGly, and GlyGlyGly) adducts can be found in MALDI [163]. In their experiment, CD–guest complexes were observed regardless of whether an aromatic moiety existed or not, and also maltoheptaose (the linear analogue of β-CD) showed a similar complexation even though it does not have any cavity to capture a guest molecule (Figure 9). In contrast, in the studies by So et al., CD–amino acids complexes represented the characteristics of inclusion complexes, such as chiral differentiation and size specificity, in MALD-MS experiments [164]. The Zenobi group pointed out that the different results may come from the different sample preparation; for example, Lehmann et al. used the dried droplet method, and a layer-by-layer method was used by So et al. [28].

As described above, these opposing views are heavily influenced by what exactly constitutes inclusion complexes when viewed in light of the CD–guest complexes observed in ESI-MS and MALDI-MS studies. Recently, more sophisticated methodologies, such as ion-mobility spectroscopy and IR multiphoton spectroscopy, have been used to conduct structural elucidation analysis of gaseous CD–guest structures. Further discussions about the basic principles governing these processes will be discussed in subsequent sections.

### 4.2. Ion-Mobility Spectrometry

Ion-mobility spectrometry (IMS) is a renowned and widely used analytical technique for separating gaseous analytes based on their collisional cross-section (CCS) [165,166,167]. IMS is often incorporated into state-of-the-art commercial mass spectrometry instruments to simplify the product matrix of the complex mixtures obtained from various biological samples before tandem mass analyses [168,169,170]. The technique’s applications include the purification of complex mixtures containing peptides, lipids, glycans, and metabolites [168,171,172], detection of explosives in chemical warfare, analysis of environmental pollutants and crude oil, and applications in the field of forensics [170,172,173,174]. Thanks to its broad-scale applicability, IMS has evolved into distinct branches, namely, drift time IMS (DTIMS), high-field asymmetric IMS (FAIMS), travelling-wave IMS (TWIMS), and trapped IMS (TIMS). The applications and governing operating principles of these techniques are detailed elsewhere [171,175]. IMS analysis provides information on molecular structures as it is capable of separating the analytes of interest according to the ion mobility directly related to their sizes and shapes. The structural elucidation of the gaseous CD complex conformers has been conducted in numerous IMS studies. In this section, we will review literature related to the analysis of CD complexes using IMS.

#### 4.2.1. Confirmation of the Formation of CD–Guest Complexes and Their Associated Conformers

An IMS/ESI-MS combined analytical technique has been used to confirm the formation of host–guest CD complexes and the existence of various conformers. Mathiron et al. utilized ESI-MS analysis to confirm the formation of α-CD/moringin (4-(α-l-rhamnopyranosyloxy)benzyl isothiocyanate) complexes. Here, maltohexaose, a linear oligosaccharide which was used as the control, produced no evidence of interaction with α-CD [176]. The IMS/ESI–MS results clearly showed the presence of multiple α-CD/moringin (1:1) complex conformers.

IMS studies conducted by Qi et al. evaluated the existence of multiple isomers in peptide/β-CD complexes [162]. As described above, ECD-MS analysis showed that the examined peptides were bound to the β-CD via specific amino acid-based hydrogen bonding interactions, in which the amino acid that was noncovalently bound to the β-CD was different peptide-to-peptide. Differential IMS studies of the β-CD/bombesin (i.e., the peptide) complexes revealed two broadly separated peaks, indicating that two or more conformers existed. This result was proof that different binding amino acids could generate different complexes with different CCSs.

#### 4.2.2. Elucidating the Structures of CD–Guest Complexes

Structural information on gaseous CD–guest complexes, including details about the stoichiometric ratio of the host:guest and their respective shapes and sizes, were obtained using IMS. One example of this is the IMS analysis of complexes formed from CD (α- and β-CD) and coumaric acid (*o*-, *m*-, and *p*-coumaric acids, CA). Here, various stoichiometric ratios of the CD/CA complexes, such as 1:1, 2:1, or 1:2, were produced and investigated. In this study, the structure of each complex was analyzed based on the respective CCS values [177]. When compared with α-CD, the CCS values for the [α-CD + CA]^+^ and [α-CD + 2CA]^+^ complexes increased 0.05% and 0.15%, respectively. These findings indicated that α-CD encapsulated one CA molecule in the [α-CD + CA]^+^ complex, but the other CA molecule in the [α-CD + 2CA]^+^ complex was excluded from the α-CD cavity due to limited space. This was attributed to the size of the α-CD cavity, which was a little bigger than the size of the CA molecule. Relative to [2CD]^+^, a smaller CCS value was observed for the [2CD + CA]^+^ complexes, implying that CA was “sandwiched” between two CDs and, thus, had adopted a more compact conformation. Sandwich-like CD complexes were also reported between β-CD and 4,4-(propane-1,3-diyl) dibenzoic acid (PDDA) or Ibuprofen (Ibu) [178]. When the CCS values were compared, an obvious change was noted during the formation of the noncovalent complexes with [2CD − 2H^+^]^2−^ or [CD – H^+^]^−^. Here, the changes in the CCS value for the noncovalent complexes with [2CD − 2H^+^]^2−^ were much less than that of [CD – H^+^]^−^, indicating the possibility of forming sandwich-like complexes such as [2CD + PDDA/Ibu – 2H^+^]^2−^.

IMS-MS was used to gather structural information on CD (i.e., α-, β-, and γ-CD) complexes with folic acid (FA) [179]. Here, IMS results showed that the CCS values of the noncovalent complexes changed following the order [α-CD + FA – 2H]^2−^ > [β-CD + FA – 2H]^2−^ > [γ-CD + FA – 2H]^2−^ when compared with the CCS values obtained for α-, β-, and γ-CD-only compounds. In particular, α-CD showed substantial changes in its CCS value after complexation with FA. It was interpreted that the FA molecules were only partially threaded through the cavity of α-CD and β-CD, particularly for the case of α-CD. Based on this assumption, the structural conformation of each complex was calculated using semi-empirical methods such as PM6 and PM7, as depicted in Figure 10.

Chen et al. focused on the complexation of three amino acids, namely, Gly, l-Leu, and l-Phe, with α- or β-CDs [26]. For the [α-CD + Gly]^−^ and [β-CD + Gly/Leu/Phe]^−^ complexes, the differences in the CCS values were less than 6% relative to their CD-only counterpart, indicating that inclusion complexes had been formed. In contrast, [α-CD + Leu]^−^ and [α-CD + Phe]^−^ showed increases of 9% and 12% in their CCS values respectively, which was clear evidence of the formation of exclusion complexes. By comparing the sizes of each amino acid and the cavity of each CD, the authors of this study showed that the cavity size of α-CD was enough to encapsulate Gly, but not Leu or Phe. These results indicated that α- and β-CD formed inclusion complexes with Gly^−^, whereas complexation of α-CD with Leu^−^ and Phe^−^ facilitated the formation of exclusion complexes. Thus, the binding of CDs with anionic amino acids was size-dependent. Structural analysis of CD/amino acid complexes conducted using molecular dynamics (MD) simulations showed similarities between the theoretical and experimental observations, indicating that the relevant bonding associations were primarily through polar interactions via hydrogen bonds. Thus, the small Gly^−^ molecule was located inside the α-CD cavity, whereas larger Leu^−^ and Phe^−^ molecules were outside (Figure 11). This was in contrast to the structures of the [permethylated β-CD + Ala/Ile]^+^ complex, which is discussed as follows: Interactions of the protonated Ala^+^ and Ile^+^ produced exclusion complexes with permethylated β-CD irrespective of the size of the protonated amino acids, with the –CO_2_H functionality of the amino acid protruding out of the CD unit. It was thought that the formation of the exclusion complex [permethylated β-CD + Ala]^+^ was due to steric hindrance between the –OCH_3_ groups and the amino acid moiety in the gaseous phase.

#### 4.2.3. Chiral Differentiation

Generally, IMS is unable to differentiate minor structural changes, particularly when the substrate is much smaller than the CD molecule. This situation was noted in the IMS analysis of complexes formed between α- or β-CDs and *o*-, *m*-, or *p*-CA. Here, the coumaric acid molecules could not be differentiated when CA was in the CD cavity, resulting in a minor (less than 6%) change in the CCS values [177]. DTIMS experiments conducted on the complexes of permethylated β-cyclodextrin (per-CD) and d-isoleucine (per-CD/d-isoleucine) exhibited a slightly smaller CCS value (0.6%) than its l-isoleucine counterpart [60].

The above-mentioned limitations were overcome by using SLIM (Structures for Lossless Ion Manipulations) and SUPER IM-MS (Serpentine Ultralong Path with Extended Routing IM-MS) platforms, as reported by Nagy et al. [180]. In this system, the length of the drift tube was extended up to 58.5 m by adopting a serpentine shape. The researchers in this study determined that all compounds tended to form 2:1 host–guest complexes, wherein 3-deoxy 3-amino α- or β-CD was utilized as the host molecule. They attributed the exclusive formation of 2:1 host–guest complexes to the flexible and unstable structural characteristics of [β-CD + H^+^]^+^, which had been revealed in the ion-mobility chromatogram. Subsequent separation of d- or l-amino acid mixtures, such as aspartic acid (Asp), threonine (Thr), and serine (Ser), complexed with α- or β-CD was achieved using only this 2:1 host–guest complex system. Here, separation was shown to be better in the α-CD complexes than in the β-CD complexes. It was also observed that the addition of metal ions (i.e., Li^+^ or K^+^) or water generally helped in distinguishing the chiral complexes due to the distortion of the CD cavity via the partial filling by additional species, thereby resulting in better selectivity.

Bian et al. investigated the chiral differentiation and complex conformations of (+)-catechin (CA) and (−)-epicatechin (EC) by IMS [181]. In this study, (+)-catechin and (−)-epicatechin could be differentiated only when they were complexed with an HP/γ-CD ((2-hydroxypropyl)/γ-cyclodextrin) complex or CD complexes, such as β-CD, HP/β-CD, and 2,6-di-*O*-methyl/β-cyclodextrin (DM/β-CD). It was also observed that each complex had two conformers generated from various catechin-based inclusion complexes, more specifically, the “A-ring included” versus the “B-ring included” conformers. The authors also demonstrated that each complex could be quantified via IMS through mathematical optimization despite the minor overlap observed between neighboring conformers.

### 4.3. IRMPD Spectroscopy

Research has shown that IRMPD spectroscopy coupled with MS detection offers valuable information about the structure of a molecule or a noncovalent complex. A brief description of IRMPD spectroscopy is as follows. First, ionized gaseous molecules or noncovalent complexes are isolated in a trapping device, such as an FT-ICR or an ion trap, wherein multiple pulses of a tunable IR laser or free-electron laser irradiate the isolated molecules or complexes. As the resonant IR photons are absorbed, the absorbed energy is redistributed among many vibrational modes. This causes the internal energy to exceed a certain threshold, leading to fragmentation of the molecule’s covalent bonds or separation of the constituent identities of the noncovalent complexes [182,183,184,185]. The IRMPD spectrum can provide detailed information on basic structural features, secondary geometry of the molecular systems, and the position of the charge or bonding interactions when used in combination with theoretical calculations [185].

IRMPD spectroscopy has provided an important tool for the structural elucidation of gaseous CD complexes containing guest molecules [56,59,60]. Permethylated β-CD (per-β-CD) complexes with H_2_O guest molecules, which is the simplest guest molecule available, was the first system to be analyzed using this spectroscopic technique [56]. Here, the hydroxyl groups of β-CD were all permethylated to simplify the resulting IR spectra and facilitate easier interpretation. Figure 12 shows its experimental IRMPD spectrum in comparison with two calculated IR spectra and the corresponding structures. As noted, the experimental IRMPD spectrum (Figure 12a) consisted of three intense absorption bands at 3096, 3315, and 3490 cm^−1^. As depicted in Figure 12c, the most stable conformer had a water molecule located inside the per-β-CD cavity in the form of a hydronium ion H_3_O^+^. However, the theoretical IR spectrum obtained using DFT B3LYP/6-31G calculations did not match the experimental spectrum obtained. Instead, another conformer shown in Figure 12d, which was calculated to be less stable than the former conformer by 22 kcal·mol^−1^ (i.e., of the Gibbs free energy), had H_2_O and H^+^ interacting with the per-β-CD unit separately. The associated theoretical spectrum closely matched the experimental IR spectrum, as shown in Figure 12b, indicating that the experimentally observed gas-phase structure of the per-β-CD/H^+^/H_2_O system corresponded to this conformer. The experimental observation of the very unstable (i.e., of unusually high Gibbs free energy) conformer (Figure 12d), rather than the thermodynamically favorable conformer (Figure 12c), was quite a surprise. Similar results were noted for gaseous structures that possessed very high Gibbs energy values such as per-β-CD/H^+^/amino acid complexes [59,60].

Two notable features were characteristic of this prototypical host–guest system. First, there were conspicuous differences in the IRMPD spectra of per-β-CD/H^+^/L-amino acid and per-β-CD/H^+^/D-amino acid complexes. Second, the structures of the gas-phase per-β-CD/H^+^/amino acid complexes elucidated by DFT calculations were much less stable (i.e., possessed much higher Gibbs free energy) than the global minimum conformers. The experimentally observed chiral differentiation of the per-β-CD/H^+^/amino acid complexes via IRMPD was identical to the results obtained for various local interactions between the per-β-CD unit and the protonated amino acid. Further experimental clarification of the latter feature, which had been observed in highly unstable conformers in the gas phase, is needed.

In Figure 13, the IRMPD spectra of the per-β-CD/H^+^/d-Ala and /l-Ala complexes are given along with the calculated IR spectra [59]. The most conspicuous features of the IRMPD spectra are the strong absorption bands near 3700 cm^−1^, which are very high-frequency absorption bands rarely observed in IR spectroscopy. The extremely blue-shifted bands at ~3700 cm^−1^ were attributed to the rare intense blue-shift of an isolated carboxyl –OH group, even though the carboxyl –OH stretch mode of the isolated gaseous Ala typically appears at 3560 cm^−1^ and the –OH hydrogen-bonded mode is usually observed at <3300 cm^−1^. Normal mode analysis of the calculated absorption bands at 3747 and 3746 cm^−1^ for the per-β-CD/H^+^/d-Ala and /l-Ala complexes respectively, indicated that this, indeed, was the case. At least for the per-β-CD/H^+^/d-Ala and /l-Ala complexes studied by Oh et al., the mode of interaction between per-β-CD and protonated Ala was unambiguous. The protonated Ala was not located inside the CD cavity, but rather tended to interact with the CD unit at the hydrophilic rim, with the carboxyl group protruded from the CD instead of towards it. Figure 14 presents similar findings for the per-β-CD/H^+^/d-Ile and /l-Ile systems [60]. Here, the high-frequency absorption bands observed at <3500 cm^−1^ in the experimental IRMPD spectra indicated that the –CO_2_H group in the complexes was isolated and was not incorporated in hydrogen bonding interactions with the CD unit.

As for the per-β-CD/H^+^/H_2_O system described above, the calculated structures of the per-β-CD/H^+^/Ala and per-β-CD/H^+^/Ile complexes were thermodynamically unfavorable (i.e., of very high Gibbs free energy) due to the lack of hydrogen bonding interactions between the –CO_2_H group of the amino acid and the per-CD unit. The Gibbs free energy of the resulting conformer was higher by 17–21 (13–14) kcal·mol^−1^ than the most stable conformer (Figure 15) of the per-β-CD/H^+^/Ala (per-β-CD/H^+^/Ile) complexes. In the latter conformers, the –CO_2_H group bent toward the per-β-CD moiety, forming strong hydrogen bonding that stabilized them. Observing these highly thermodynamically unstable per-β-CD/H^+^/Ala and per-β-CD/H^+^/Ile complexes in the gas phase is extremely riveting, and their origin remains unclear. Oh, Lee, and colleagues proposed several possible explanations to justify these findings. First, it was speculated that a higher energy conformer may have been kinetically trapped before relaxing to the global minimum conformer while a mixture of per-CD, H^+^, and the amino acid analyte in solution were being transferred into the gas phase [186,187,188,189]. In other words, the protonated amino acids interacted with numerous water molecules outside the per-CD cavity in the solution, and this conformer could not be converted to the global minimum conformer in the gas phase. Second, it may be that different ion-optics conditions could induce different conformer populations, thereby leading to spectral differences [190,191,192].

Altogether, IRMPD studies combined with DFT calculations have revealed that gaseous CD–guest noncovalent complexes were not inclusion complexes, in contrast to the complexes formed in the solution.

## 5. Conclusions

The host–guest chemistry of CDs is of keen interest for a variety of commercial and academic applications because CDs can encapsulate the guest molecule to allow versatile applications, including the separation of chiral molecule as well as the improvement of solubility/stability/biocompatibility, and reduction of toxicity/pollution of guest molecules. These versatile and interesting features of CDs in the solution phase have been extensively studied by NMR spectroscopy and other methodologies, focusing on the structural characteristics. As a result, many researchers presented that CDs in the solution phase formed the inclusion complex by encapsulating a guest molecule via hydrophobic interactions in the CD cavity. However, the structure of gaseous CD–guest noncovalent complexes remains unclear. Until now, the three different structures have been suggested, including inclusion complex, partial inclusion complex, and electrostatic adducts. The deviation from the solution phase structure may come from the condition of ESI and also from the difference of main driving forces for the formation of noncovalent complexes in the solution and gas phases. Recently, new MS techniques have been applied to better understand the essence of the host–guest chemistry in the gas phase, including ESI-mass spectrometry experiments, ion-mobility measurements, IRMPD spectroscopy, and density functional theory (DFT) calculations. These efforts have deepened our understanding of the structure of noncovalent CD–guest complexes, and a better understanding will be achieved with the advent of newer technologies. We would like to conclude that more rigorous validation studies are needed in the future to obtain an accurate picture of gaseous noncovalent CD–guest complexes.

## Figures and Tables

**Figure 1 molecules-25-04048-f001:**
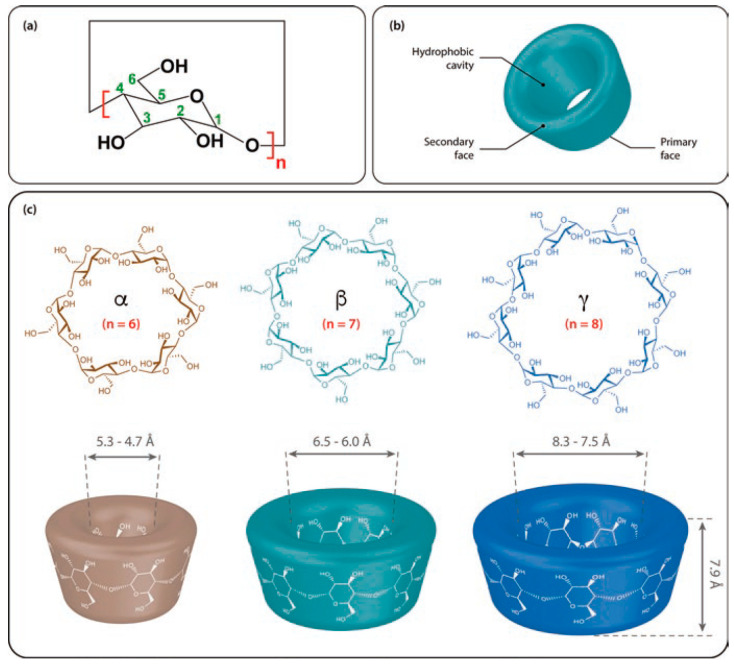
Cyclodextrin (CD) structures of (**a**) the monomer unit, (**b**) the overall shape, and (**c**) the structures of α-, β-, and γ-CDs. Reprinted with permission from Springer Nature: Springer International Publishing AG, Cyclodextrin Fundamentals, Reactivity and Analysis by Crini et al., Reference [3], COPYRIGHT (2018).

**Figure 2 molecules-25-04048-f002:**
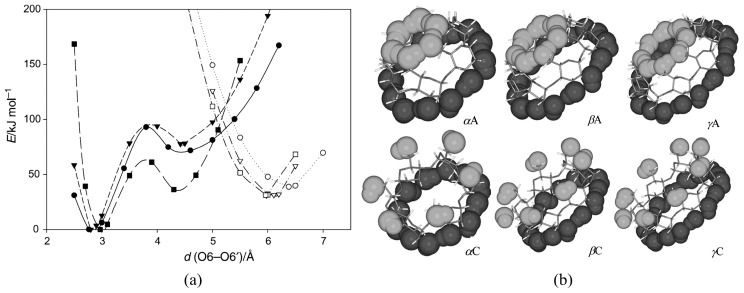
(**a**) Energy profiles of α(circle)-, β(triangle)-, and γ (rectangle)-CDs obtained by varying the O6–O6′ distances in the primary hydroxyl groups. The filled and unfilled symbols denote conformers from the A and C series, respectively. (**b**) A and C conformations of α-, β-, and γ-CDs. Reprinted with permission from Springer Nature, Springer-Verlag, Monatsh Chem., Reference [127], COPYRIGHT (2008).

**Figure 3 molecules-25-04048-f003:**
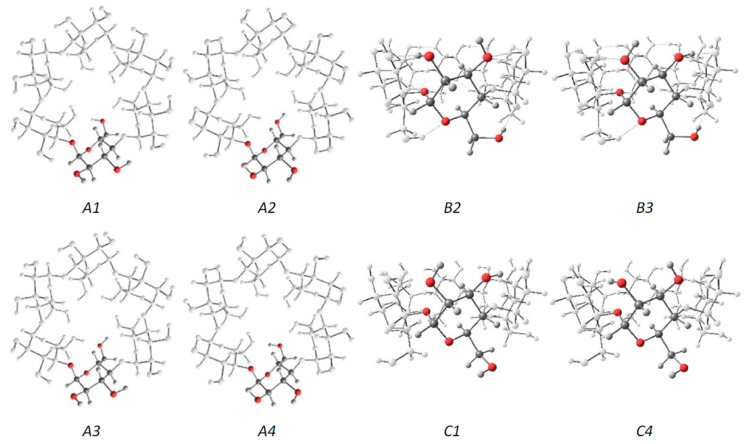
The three series of α-CD conformers. Each glucose unit is filled in for easier visualization. Reprinted with permission from Springer Nature, Springer-Verlag, J. Mol. Model. Reference [128], COPYRIGHT (2008).

**Figure 4 molecules-25-04048-f004:**
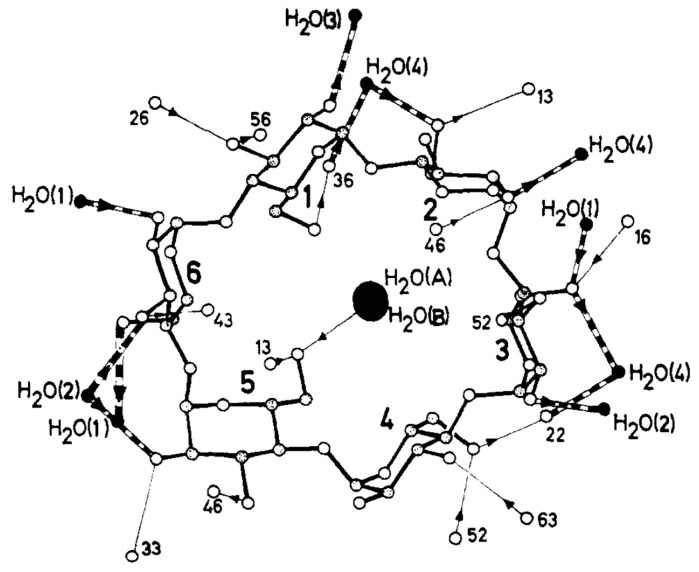
Structure of α-CD·(H_2_O)_6_ and the hydrogen bonding network located inside the α-CD cavity. Reprinted with permission from Reference [132]. Copyright (1974) American Chemical Society.

**Figure 5 molecules-25-04048-f005:**
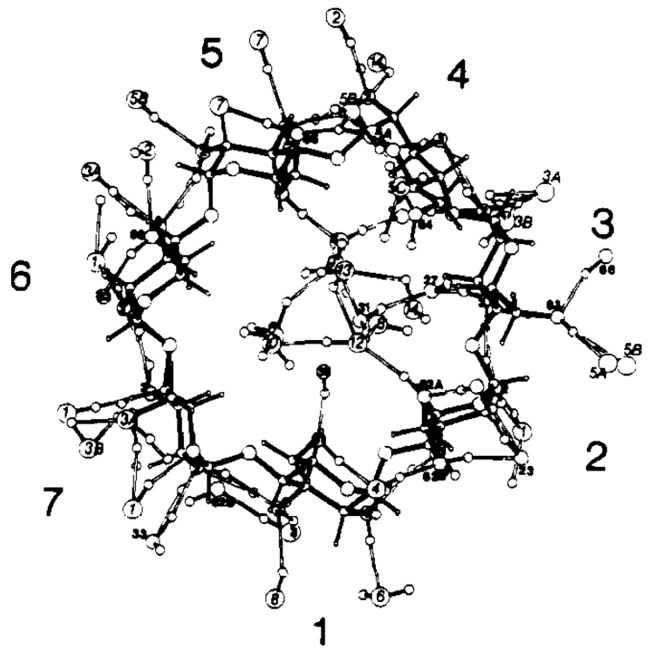
Structure of β-CD·(H_2_O)_11_ and the associated hydrogen bonding networks inside the β-CD cavity, as determined via neutron diffraction. Reprinted with permission from Reference [136]. Copyright (1984) American Chemical Society.

**Figure 6 molecules-25-04048-f006:**
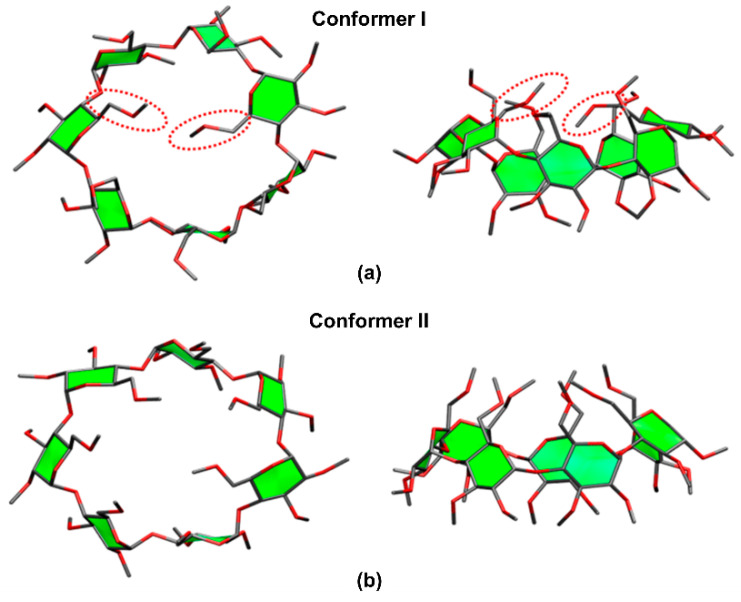
The most stable conformers of permethylated β-CD. Reproduced from Reference [56] by permission of the PCCP Owner Societies.

**Figure 7 molecules-25-04048-f007:**
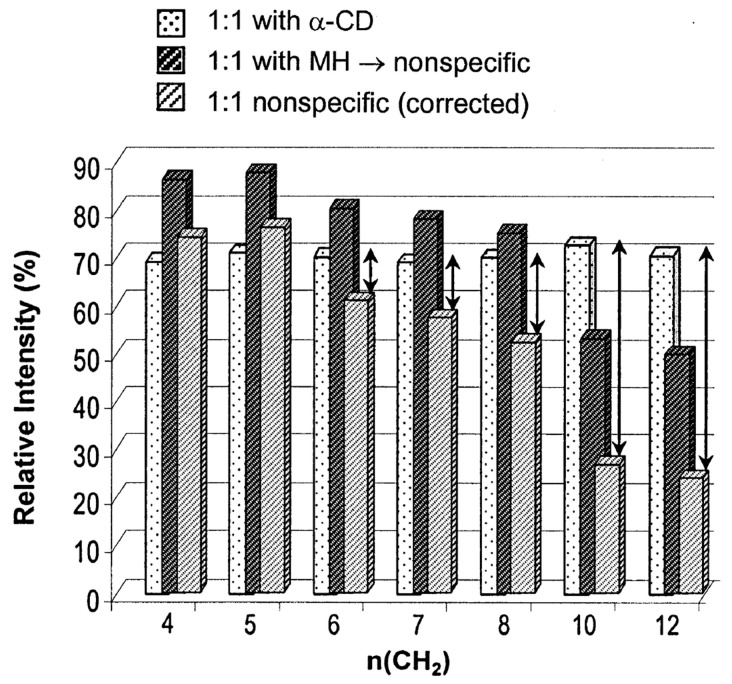
Relative intensities of the α-CD/diacid complexes (white bar with dots), the maltohexaose/diacid complexes (blank bar with dark hatches), and the maltohexaose/diacid complexes after the correction using the response factor (grey bar with light hatches). The arrows correspond to the contribution of inclusion complexes, assuming the nonspecific adducts in both complexes contribute equally. Reprinted from Reference [155], Copyright (2002), with permission from Elsevier.

**Figure 8 molecules-25-04048-f008:**
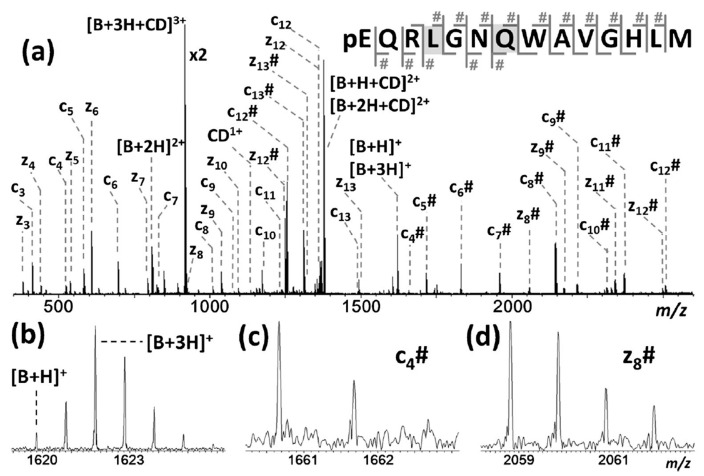
(**a**) Electron capture dissociation (ECD) mass spectrum of the triple-charged β-CD^+^ bombesin peptide ion along with the peptide cleavage map. Multiple c/z fragments were detected and β-CD binding fragments were marked with #. (**b**) The generation of unusual [peptide + 3H]^+^ ions, meaning the hydrogen transfer from the CD to the peptide at the hydrogen bonding position. (**c**,**d**) The mass spectra of c_4_# and z_8_# ions corresponding to the CD binding site. Reprinted with permission from Reference [162]. Copyright (2015), American Chemical Society.

**Figure 9 molecules-25-04048-f009:**
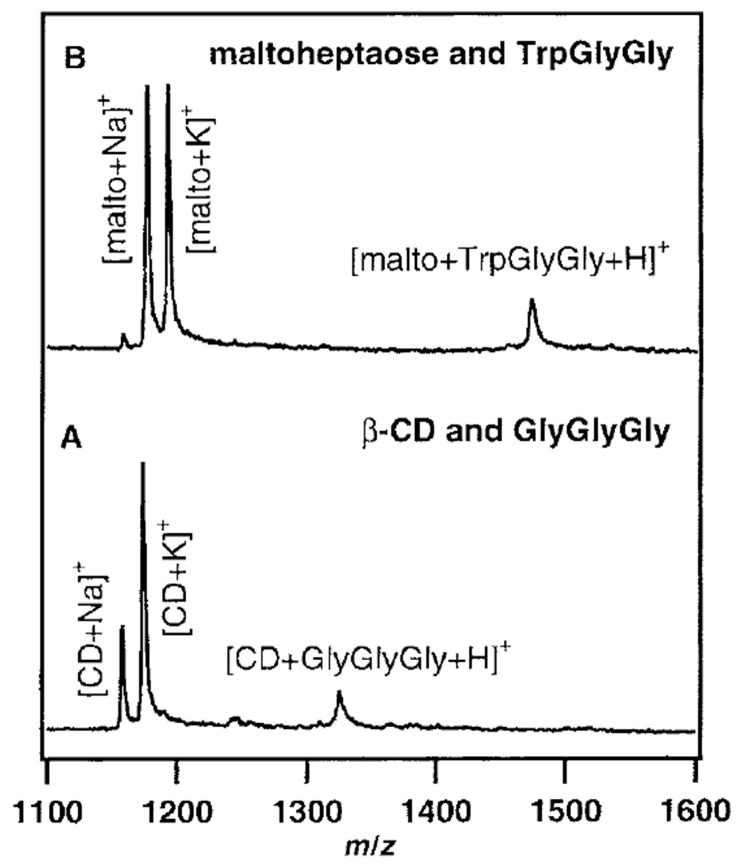
MALDI-MS spectra of (**A**) β-CD and GlyGlyGly, 1 + 1 molar ratio, and (**B**) maltoheptaose and TrpGlyGly, 1 + 1 molar ratio. Reproduced from Reference [163] by permission of the PCCP Owner Societies.

**Figure 10 molecules-25-04048-f010:**
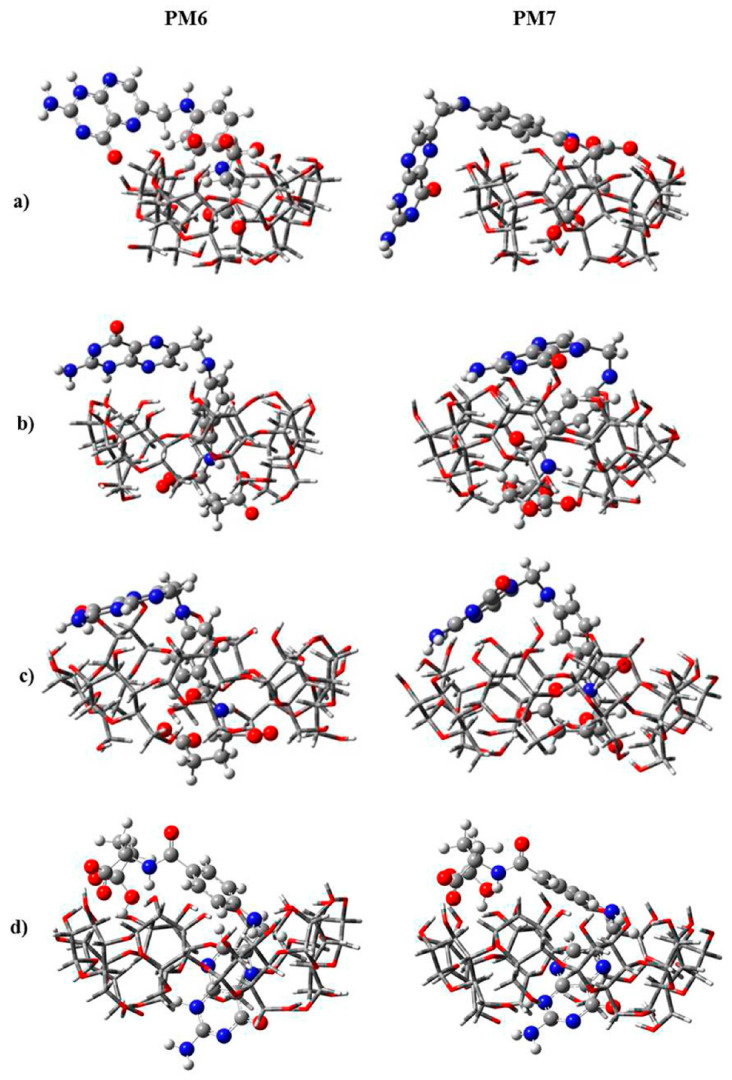
Structures of the most energetically favorable doubly negatively charged noncovalent complexes of FA with (**a**) α-CD, (**b**) β-CD, and (**c**,**d**) γ-CD obtained from PM6 and PM7 semiempirical calculations. Reprinted with permission from Reference [179]. Copyright (2014), American Chemical Society.

**Figure 11 molecules-25-04048-f011:**
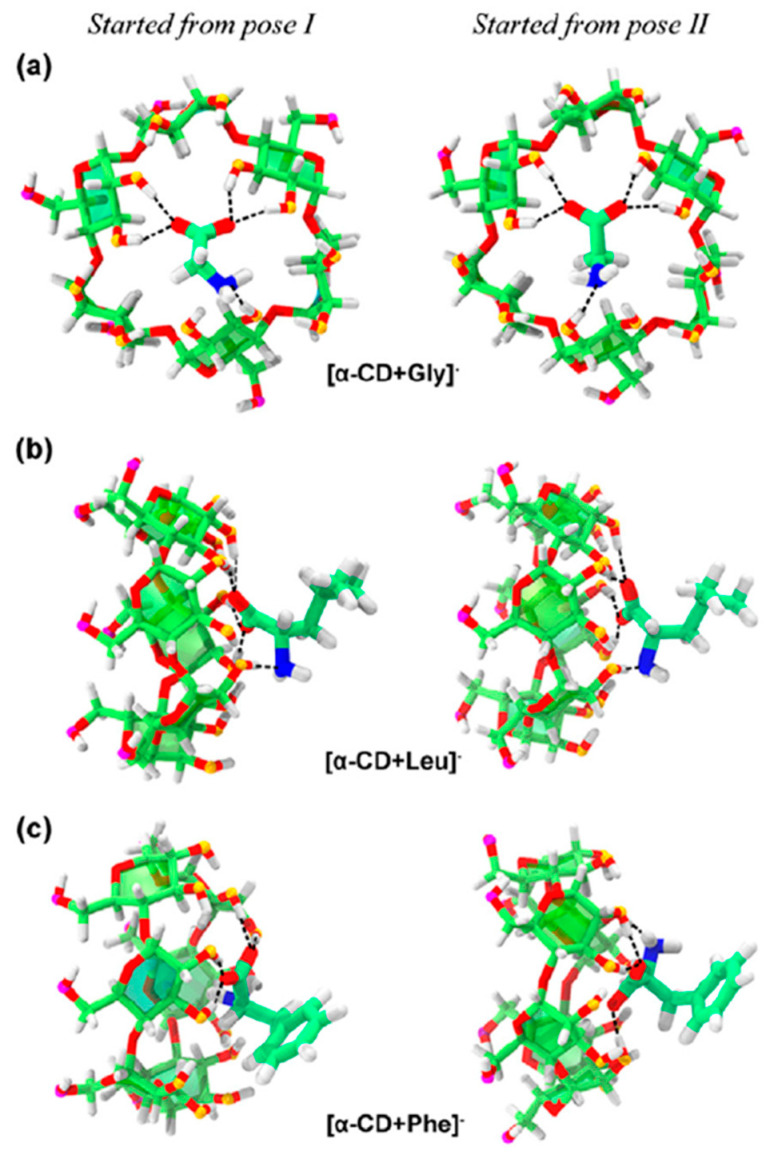
Structures of (**a**) [α-CD + Gly]^−^, (**b**) [α-CD + Leu]^−^, and (**c**) [α-CD + Phe]^−^ complexes obtained via MD simulations. Reprinted from Reference [26], Copyright (2018), with permission from Elsevier.

**Figure 12 molecules-25-04048-f012:**
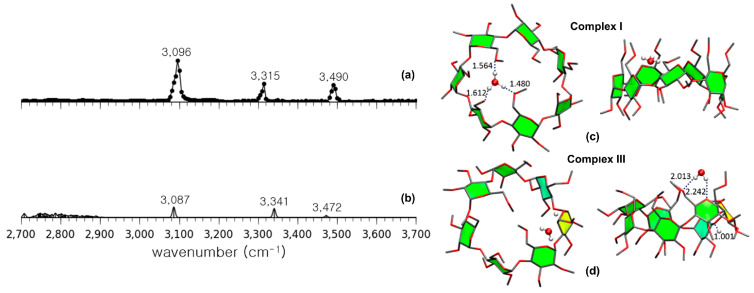
(**a**) The experimentally obtained IR photodissociation spectrum of permethylated β-CD–H_3_O^+^ (Complex I) and (**b**) the calculated spectrum of permethylated βI-CD–H^+^/H_2_O (Complex III). (**c**) Structures of Complex I and (**d**) Complex III. Reproduced from Reference [56] by permission of the PCCP Owner Societies.

**Figure 13 molecules-25-04048-f013:**
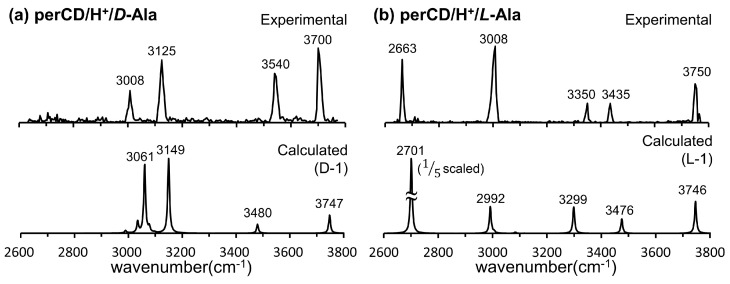
Experimental IRMPD and calculated IR spectra of the per-β-CD/H^+^/d-Ala and /l-Ala complexes. Reproduced from Reference [59] by permission of the PCCP Owner Societies.

**Figure 14 molecules-25-04048-f014:**
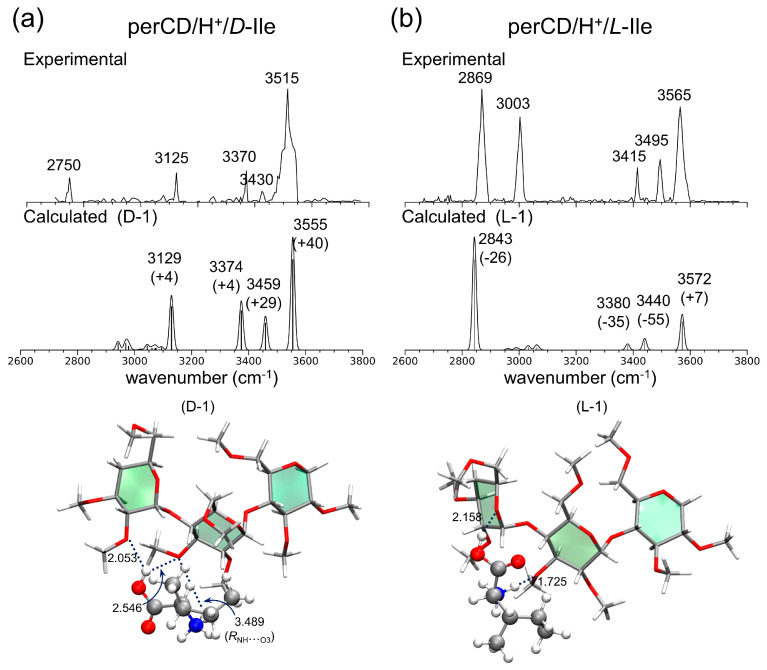
Experimental IRMPD and calculated IR spectra of (**a**) per-β-CD/H^+^/D-Ile and (**b**) /L-Ile complexes with the associated structures. Reproduced from Reference [60] by permission of the PCCP Owner Societies.

**Figure 15 molecules-25-04048-f015:**
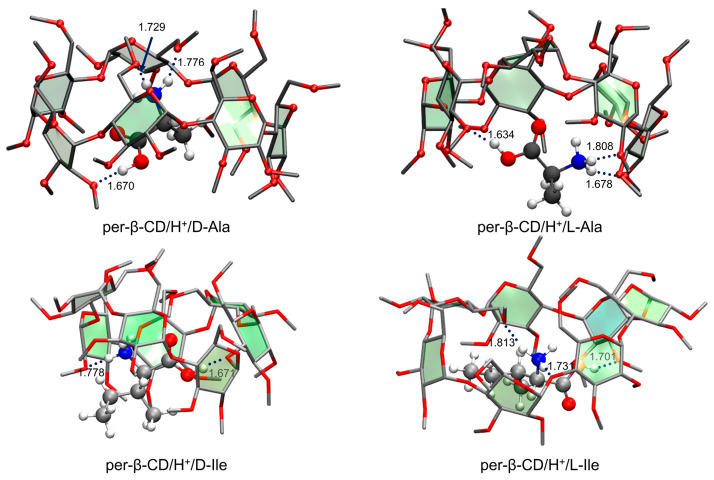
The most stable conformers of per-β-CD/H^+^/Ala and /Ile complexes. Reproduced from Reference [59,60] by permission of the PCCP Owner Societies.

**Table 1 molecules-25-04048-t001:** The three native cyclodextrins and their main characteristics. Reprinted with permission from Springer Nature: Springer International Publishing AG, Cyclodextrin Fundamentals, Reactivity and Analysis by Crini et al., Reference [3], COPYRIGHT (2018).

Cyclodextrin	α	β	γ
Chemical Abstracts Service Registry Number	100016-20-3	7585-39-9	17465-86-0
Glucopyranose units	6	7	8
Formulae	C_36_H_60_O_30_	C_42_H_70_O_35_	C_48_H_80_O_40_
Molecular weight (g/mol)	972.9	1135.0	1297.1
Central cavity diameter: external/internal (Å)	5.3/4.7	6.5/6.0	8.3/7.5
Height of torus (Å)	7.9	7.9	7.9
Approximate volume of cavity (Å)	174	262	427
Water solubility at 25 °C (g/L)	145	18.5	232
Number of water molecules within cavity	6–8	11–12	13–17
pKa	12.3	12.2	12.1

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
