# Peer review of "Noncovalent Complexes of Cyclodextrin with Small Organic Molecules: Applications and Insights into Host–Guest Interactions in the Gas Phase and Condensed Phase"

_molecules, 2020, doi:10.3390/molecules25184048_

Round 1
Reviewer 1 Report
The review about “Interactions of cyclodextrins with guest molecules in 2the gas phase: Experiments and Theoretical Studies” is very interesting and well carried out. The reviewer considered that the manuscript was well-written and contained important relevant studies; indeed, point 3 is very interesting. Therefore, the reviewer believed that this review would be contributing to the development of works about CD-gas interactions.
However, the review lacks information about (essential oil or volatile)/CD interactions. They are able to separate them and optimize the release. As the work with CD in fragrances is very important int the industry, this part needs to be included in this review (e.g. as point 2.6). Some references to illustrate: https://doi.org/10.1016/j.foodchem.2015.10.023 ; https://doi.org/10.1016/j.chemphyslip.2019.02.001 ; 10.1007/s10847-009-9726-3 ; https://doi.org/10.1002/pca.2654 ; https://doi.org/10.1007/s10847-017-0744-2
I recommend a major revision where authors introduce this part in the review.
Author Response
Reviewer 1
The review about “Interactions of cyclodextrins with guest molecules in the gas phase: Experiments and Theoretical Studies” is very interesting and well carried out. The reviewer considered that the manuscript was well-written and contained important relevant studies; indeed, point 3 is very interesting. Therefore, the reviewer believed that this review would be contributing to the development of works about CD-gas interactions.
[Q1] However, the review lacks information about (essential oil or volatile)/CD interactions. They are able to separate them and optimize the release. As the work with CD in fragrances is very important in the industry, this part needs to be included in this review (e.g. as point 2.6). Some references to illustrate:
https://doi.org/10.1016/j.foodchem.2015.10.023;
https://doi.org/10.1016/j.chemphyslip.2019.02.001; citronellar
10.1007/s10847-009-9726-3; x
https://doi.org/10.1002/pca.2654 ; separation science
https://doi.org/10.1007/s10847-017-0744-2 review paper 2017
[Response] We appreciate the reviewer for the favorable comments. As the reviewer suggested, we added the applications of the essential oils as follows. The added sentences are denoted in red.
Line 168-186:
2.6. Essential oils
Essential oils are terpene and terpenoids-like volatile compounds presenting strong odors, which are generated from the aromatic plant. These oils have many uses in the fields of medicals, foods, cosmetics, and therapeutics because of antifungal/antimicrobial/antioxidant functions, fine flavor, and analgesic/sedative properties. However, the oils have drawbacks of unfamiliar characteristics such as volatility, poor solubility, and poor stability against heat/light/oxidation. CDs have become a useful strategy to overcome these challenges by enhancing physicochemical properties using inclusion complexes. A wide variety of applications using CD complexes with essential oils and volatiles are recently reviewed [117,118]. Still, the extensive researches of CD/essential oil complexes have been conducted to improve the properties of essential oil [119–124]. For example, Abril- Sánchez et al. presented the improved efficacy of the essential oil citronellal by encapsulating it using HP-β-CD [119]. As expected, the encapsulation increased solubility, and the enhanced durability was also observed by GC/MS and sensory analysis. Additionally, when citronellal, HP-β-CD, and Glucobay® (an antidiabetic drug) were mixed to prevent the growth of antimicrobial, the synergetic effect was observed with a long-lasting effect. The CD nanosponge (defined as CD polymers) /cinnamon oil complex was also evaluated as food packaging components having a bactericidal effect [121]. In the referred paper, CD nanosponge/cinnamon oil complex showed a better antibacterial effect than cinnamon oil alone even at a lower amount with the increased stability.
Reviewer 2 Report
The proposed manuscript of a review on the host-guest chemistry of cyclodextrins (CDs) in the gas phase constitutes a well-elaborated, comprehensive and insightful account on this important subject of supramolecular chemistry. The referenced articles seem to have been been chosen carefully and with good understanding of the topic. The introduction provides optimum amount of information that both the readers new to the topic as well as advanced supramolecular researchers can find useful. Structural aspects are described in a precise but understandable manner. I particularly like how the authors describe different approaches taken by the researchers towards investigating the supramolecular interactions of CDs. It is well emphasized that the interactions in the gas phase may differ much from the ones in liquid and solid state. The topic of investigating the supramolecular chemistry of CDs in gas phase is exciting and vibrant, the concepts of the formation of inclusion complexes vs non-covalent systems are presented critically, analytical techniques are introduced and the results discussed in a very skillful manner.
The style, language and editing sides of the manuscript are very strong. I would like to propose only several minor editing changes to be made:
line 76: "interactions of gaseous CDs" -> "interactions of CDs in the gas phase"
line e.g. 221, 223, 235, 240: subscripts needed for the numbers in H2O, OCH3 etc. (please check throughout the manuscript)
line 262: "specifity" -> please consider whether the term "selectivity" wouldn't be better in this case
line 289, 290, 301: it might be a good idea to provide the reader with what the numbers given (1.6, 3.1 and so on) exactly mean
line 363: "maltohexaose" instead of "matohexaose"
line 479: "i.e., of unusually high Gibbs free energy" (and, similarily, add "of" in all places were Gibbs free energy is mentioned in the context of the molecules' stability)
line 507: "with the carboxyl group protruded toward the vacuum instead of toward the CD" -> "with the carboxyl group protruded from the CD instead of towards it"
However, I'd like to underline that the abovementioned corrections are proposed only to improve certain fragments of the manuscript, and do not affect my overall very high rating of the manuscript's quality.
To sum up, I fully recommend the manuscript to be published in "Molecules". It corresponds well with the Journal's scope and I'm pretty sure will be useful to a wide readership.
Author Response
Reviewer 2
The proposed manuscript of a review on the host-guest chemistry of cyclodextrins (CDs) in the gas phase constitutes a well-elaborated, comprehensive and insightful account on this important subject of supramolecular chemistry. The referenced articles seem to have been chosen carefully and with good understanding of the topic. The introduction provides optimum amount of information that both the readers new to the topic as well as advanced supramolecular researchers can find useful. Structural aspects are described in a precise but understandable manner. I particularly like how the authors describe different approaches taken by the researchers towards investigating the supramolecular interactions of CDs. It is well emphasized that the interactions in the gas phase may differ much from the ones in liquid and solid state. The topic of investigating the supramolecular chemistry of CDs in gas phase is exciting and vibrant, the concepts of the formation of inclusion complexes vs non-covalent systems are presented critically, analytical techniques are introduced and the results discussed in a very skillful manner.
The style, language and editing sides of the manuscript are very strong. I would like to propose only several minor editing changes to be made:
[Q1] line 76: "interactions of gaseous CDs" -> "interactions of CDs in the gas phase"
[Response] Thank you very much for your kindness to point out minor mistakes. It is corrected.
[Q2] line e.g. 221, 223, 235, 240: subscripts needed for the numbers in H2O, OCH3 etc. (please check throughout the manuscript)
[Response] They are all corrected.
[Q3] line 262: "specifity" -> please consider whether the term "selectivity" wouldn't be better in this case.
[Response] It is corrected.
[Q4] line 289, 290, 301: it might be a good idea to provide the reader with what the numbers given (1.6, 3.1 and so on) exactly mean.
[Response] Thank you very much for your valuable advice. The detailed explanation is given in the revised manuscript as follows.
Line 318-327:
“In this experiment, the complexes trapped in the analyzer were allowed to react with alkylamines that had leaked into the analyzer, and the rate constant of the guest molecule exchange process was obtained for each complex by applying pseudo-first-order rate reaction kinetics; ln I/I0 versus t plots are used to get the slope, namely the rate constants of the complexes, where I is the sum of intensities of the CD/amino acid and CD/alkylamine at time t and I0 is the intensity of CD/amino acid at time t0. Next, the enantioselectivity of CDs for chiral amino acids was determined by calculating the ratio of the rate constants of the enantiomeric complexes, i.e, kL/kD. Here, the increase in the enantioselectivity of the complex was dependent on the amino acid side‑chain and followed the order: alanine (1.6) < valine (3.1) < leucine (3.6) ≤ isoleucine (3.8).”
[Q5] line 363: "maltohexaose" instead of "matohexaose".
[Response] It is corrected.
[Q6] line 479: "i.e., of unusually high Gibbs free energy" (and, similarily, add "of" in all places were Gibbs free energy is mentioned in the context of the molecules' stability)
[Response] They are all corrected.
[Q7] line 507: "with the carboxyl group protruded toward the vacuum instead of toward the CD" -> "with the carboxyl group protruded from the CD instead of towards it"
[Response] It is corrected.
Reviewer 3 Report
The review of Lee et al. covers the topic of interaction of cyclodextrin molecular hosts and their derivatives in the gas phase. Recent developments in the field of soft ionization methods in MS have opened broad opportunities for the investigation of the weakly bound complexes in the gas phase; therefore, this topic is very actual and should be interesting for a broad readership of chemical scientists. Still, in its current state, the review is not ripe for the publication.
- The title of the review, “Interactions of cyclodextrins with guest molecules in the gas phase: Experiments and Theoretical Studies” implies the coverage of the topic in the whole breadth. Still, the authors narrow the discussion to simple cyclodextrin complexes with H2O and amino acids, with particular focus on IRMPD measurements of their own. Other compound classes are mentioned either only briefly (e.g. coumaric acids, some pharmaceuticals) or not at all. Thus, a huge chunk of recent research results remains untouched, for example, mass spectrometric studies of CD with metal complexes, polyoxametallates, etc.
- In general, a number of seminal reviews related to the investigation of weak interactions in the gas phase using mass-spectrometric methods have not been mentioned, e.g. of M. Przybylski (Angew. Chem Int. Ed. Engl 1996. 35, 806), R. Zenobi (International Journal of Mass Spectrometry 216 (2002) 1–27) and C. A. Schalley (Mass Spectrometry Reviews, 2001, 20, 253).
- The second chapter, Applications, is totally out of point: it is completely unrelated to the MS studies of CD complexes in the gas phase. A chapter like this might be useful in a review on complexing properties of CDs in general, but not in the context of this particular paper.
- The review contains too few illustrations. In particular, sub-chapters 4.1-4.2, which should be the “backbone” for a review with such a title, have only one illustration. On another hand, there are several illustrations in the chapter on the structures of CDs and their complexes in the bulk phase, as well as several illustrations in the chapter 4.3 on IRMPD spectroscopy, albeit all based on the author’s publications. With so few illustrations it is almost impossible to get a feeling about what can be achieved by MS studies of CD complexes.
- Much more can be added to the discussion of MS fragmentation studies used for the elucidation of CD Host-Guest complex structures in the gas phase.
- The authors often refer to the “three point rule” of binding (line 57) and then apply it to the discussion of CD-amino acid complexes, but they never give a detailed explanation or an illustration to clarify this model. Only upon further reading of the review a reader can get a feeling on what it is about. The authors should explain this rule and discuss its applicability to different types of the complexes (is it applicable only to amino acids or, in general, to all possible guests?) after it was first mentioned in the paper.
To sum up, the topic is actual and such a review will likely be well accepted by the journal readership, but its current version should be thoroughly overworked and expended to allow its publication.
Author Response
Responses to the reviewers’ comments
Reviewer 3
The review of Lee et al. covers the topic of interaction of cyclodextrin molecular hosts and their derivatives in the gas phase. Recent developments in the field of soft ionization methods in MS have opened broad opportunities for the investigation of the weakly bound complexes in the gas phase; therefore, this topic is very actual and should be interesting for a broad readership of chemical scientists. Still, in its current state, the review is not ripe for the publication.
[Q1] The title of the review, “Interactions of cyclodextrins with guest molecules in the gas phase: Experiments and Theoretical Studies” implies the coverage of the topic in the whole breadth. Still, the authors narrow the discussion to simple cyclodextrin complexes with H2O and amino acids, with particular focus on IRMPD measurements of their own. Other compound classes are mentioned either only briefly (e.g. coumaric acids, some pharmaceuticals) or not at all. Thus, a huge chunk of recent research results remains untouched, for example, mass spectrometric studies of CD with metal complexes, polyoxametallates, etc.
[Response] We agree with your comment about the coverage of this review. We changed the title to “Interactions of cyclodextrins with small organic molecules in the gas phase: Experiments and Theoretical Studies” to indicate the scope of our coverage. We also added some information about other researches about metal cations, polyoxometalates, dodecaborates, and octahedral rhenium clusters in the introduction part to let readers understand the coverage of this paper and guide other untouched research fields.
The following is the added part in the introduction.
Line 52-59:
“In attempts to elucidate the specific mechanism governing CD recognition in the gas phase, researchers have noted that the three-point interaction model can be applied putatively as used in the case of the solution phase, even though there is no exact model to explain all different cases of CD-guest complex [36]. On the other hand, it is well understood about the interaction of the noncovalent complex of CD/inorganic guest molecules including metal cations [37,38], e.g., polyoxometalates [39–41], dodecaborates [42–44], and octahedral rhenium clusters [45, 46] in the gas phase. Due to the lack of hydrophobic part and repetitive structure except for metal cations, these molecules form multiple hydrophilic interaction bridges along the CDs’ exterior hydrophilic rims.”
[Q2] In general, a number of seminal reviews related to the investigation of weak interactions in the gas phase using mass-spectrometric methods have not been mentioned, e.g. of M. Przybylski (Angew. Chem Int. Ed. Engl 1996. 35, 806), R. Zenobi (International Journal of Mass Spectrometry 216 (2002) 1–27) and C. A. Schalley (Mass Spectrometry Reviews, 2001, 20, 253).
[Response] Thank you for your valuable suggestion. Reflecting your suggestion, the following paragraph is added.
Line 22-29:
“Molecules in the gas phase undergo predominantly polar interactions such as hydrogen bonding and electrostatic interactions [26] because the solvent shielding by water molecules are absent. On the other hand, the hydrophobic interaction gets removed in the gas phase unlike in the condensed phase where water molecules force the nonpolar molecules to gather each other to minimize the contact towards the polar surface [27,28]. These changes in interaction forces imply that the direct comparison of noncovalent interactions (or complexes) between the gas and condensed phases should be cautious, in particular for the quantitative determination [27–29].”
[Q3] The second chapter, Applications, is totally out of point: it is completely unrelated to the MS studies of CD complexes in the gas phase. A chapter like this might be useful in a review on complexing properties of CDs in general, but not in the context of this particular paper.
[Response] We understand the reviewer’s concern and are very thankful to the kind advice. Somehow, the other reviewer liked this part (Reviewer 2: “The introduction provides optimum amount of information that both the readers new to the topic as well as advanced supramolecular researchers can find useful.”) and asked us to add a little more about essential oils. Indeed, the applications using CDs are related to the noncovalent interactions in the solid/solution phase. To clarify this fact, we revised the subject heading to the “Applications of CDs inclusion complexes in the solid/solution phases”. Additionally, some explanation about the reason for the presentation of applications was written in the introduction part of this subject as follows.
Line 84-87
“In this section, we have opted to represent state-of-the-art industrial applications of CDs to emphasize the importance of CDs for the above-mentioned fields, in particular in the solid/solution phases, implying the meaningfulness of precise characterization of CD complex structures in the gas phase too.”
[Q4] The review contains too few illustrations. In particular, sub-chapters 4.1-4.2, which should be the “backbone” for a review with such a title, have only one illustration. On another hand, there are several illustrations in the chapter on the structures of CDs and their complexes in the bulk phase, as well as several illustrations in the chapter 4.3 on IRMPD spectroscopy, albeit all based on the author’s publications. With so few illustrations it is almost impossible to get a feeling about what can be achieved by MS studies of CD complexes.
[Response] Thank you very much for the reviewer’s insightful suggestion. We added more illustrations; Figure 7, Figure 8, Figure 9, and Figure 10.
[Q5] Much more can be added to the discussion of MS fragmentation studies used for the elucidation of CD Host-Guest complex structures in the gas phase.
[Response] Thank you very much for the valuable suggestion. We added three paragraphs on MS fragmentation studies used for the elucidation of CD Host-Guest complex structures in the gas phase.
The followings are added.
Line 289-303
“Gabelica et al. also suggested the formation of nonspecific adducts of α-CD/linear α,ω-dicarboxylic acids (diacid) in the gas phase [152]. It is well known that the association constant of this complex in the solution phase is linearly correlated with the chain length (hydrophobicity) of the diacid, verifying the formation of inclusion complex using hydrophobic interaction [152,153]. However, the mass spectrometry studies varying the chain length showed a discrepancy from this trend: 1) in the MS spectra, almost constant intensities of 1:1 complexes were observed, 2) in the survival yield curve in tandem mass spectrometry, the shorter chain length of diacid showed better stability. They suggested these results came from the contribution of electrostatic adducts. On the other hand, α-CD/diacid complexes showed stronger interaction compared to maltohexaose (the linear analog of β-CD, which necessarily brings only nonspecific complex)/diacid complexes), implying another contribution for the formation of α-CD/diacid complexes. Some seemingly inconsistent results were explained by the interplay of inclusion complexes and nonspecific adduct formation. Using the results of maltohexaose, they calculated each portion of inclusion complexes and nonspecific adducts concerning to the chain length (Figure 7) [154,155].”
Line 343-354
“Likewise, the reflection of the solution phase to the gas phase of CDs non-covalent complexes was presented by Guo et al [162]. They investigated the noncovalent complexes of α-, β-, and γ-CD/rutin of the gas phase using competition experiments, dilution test, and CID experiments in an ion trap. The binding competition between three CDs and rutin showed different binding constants, but also it was observed these values are corresponding to the ones of the solution phases. In dilution tests, the peak intensities of each complex didn’t change much following dilution, implying the contribution of nonspecific binding is small. CID experiments also showed corresponding stability of each complex to the binding constants in the solution phase. Interestingly, the fragmentation pattern showed the selective cleavage between quercetin and rutinosyl moieties, convincing the encapsulation of quercetin moiety in the hydrophobic cavity of CDs. The fragmentation of γ-CD/rutin showed the breakage of noncovalent interaction between host and guest without internal cleavage resulted from the inefficient binding due to size difference.”
Line 387-397:
“The disagreement about the structures of CD-guest noncovalent complex in the gas phase can be also found in the MALDI-MS experiments. Lehmann et al. presented that the noncovalent CD–guest (PheGlyGly, TrpGlyGly, and GlyGlyGly) adducts can be found in MALDI-MS [164]. In their experiment, CD–guest complexes were observed regardless of whether an aromatic moiety exists or not, and also maltoheptaose (the linear analogue of β-CD) showed the similar complexation even though it doesn’t have any cavity to capture a guest molecule (Figure 9). In contrast, in the studies by So et al., CD–amino acids complexes represented the characteristics of inclusion complexes, such as chiral differentiation and size specificity, in MALD-MS experiments [165]. The Zenobi group pointed that the different results may come from the different sample preparation; for example, Lehmann et al. used PNA dried droplet method, and on the other hand a layer-by-layer method was used by So et al. [166].”
[Q6] The authors often refer to the “three point rule” of binding (line 57) and then apply it to the discussion of CD-amino acid complexes, but they never give a detailed explanation or an illustration to clarify this model. Only upon further reading of the review a reader can get a feeling on what it is about. The authors should explain this rule and discuss its applicability to different types of the complexes (is it applicable only to amino acids or, in general, to all possible guests?) after it was first mentioned in the paper.
[Response] We appreciate the reviewer for the excellent suggestion about “three points rule”. In the introduction part, we added more detailed information about “three points rule” and “lock and key model”.
Line 34-43:
“The so-called “three points rule” or “lock and key model” has been applied to elucidate the mechanism governing CD recognition in solutions like other host-guest chemistry. In the three points rule, the three points generally mean one hydrophobic point and the other two interaction points (including hydrogen bonding, Coulomb interaction, and van der Waals repulsion) when it comes to the CD complex system [24,30]. For example, the complexation of CD with aromatic amino acids can be explained by the three points rule because aromatic amino acids have one hydrophobic point in the side chain and two hydrogen bonding points of CO2- and NH3+ groups. The lock and key model, which was first suggested by Emil Fisher, attributes the specific binding of enzyme and substrate to the optimal geometric fit between them [31]. In the CD complex system, this model is especially useful to explain the recognition of guest molecules lacking notable interaction parts [24].”
To sum up, the topic is actual and such a review will likely be well accepted by the journal readership, but its current version should be thoroughly overworked and expended to allow its publication.
Round 2
Reviewer 1 Report
The present manuscript is suitable for publication
Author Response
Thank you for your acception.
Reviewer 3 Report
After the author’s updates the content review looks much more balanced. Still, the major unbalance remains between the title - abstract and the contents of the review. Actually, it was the main reason of my criticism of the first version of the review: after reading of the title and the abstract, I expected to see a review focused almost exclusively on the MS studies, but the MS-related part of it started only on the page 8, which was almost half of the text of the review.
In my opinion, there is nothing wrong about having a chapter regarding the applications and studies of complex binding in the liquid/solid phase, but it should be reflected in the title and abstract of the review. My suggestions would be to announce other aspects of the review, namely, comparison of the condensed phase structures with the results obtained from the MS investigations, in the review abstract; currently the abstract it briefly mentions the applications and then almost completely focused on the MS studies. In addition, I would suggest changing the title from:
“Interactions of cyclodextrins with small organic molecules in the gas phase: Experiments and Theoretical Studies”
to something like (just a suggestion):
“Noncovalent complexes of cyclodextrin with small organic molecules: applications and insights into host-guest interactions {based on the comparative analysis of binding} in the gas phase and condensed phase”
or
“Noncovalent complexes of cyclodextrin with small organic molecules: applications and comparative analysis of host-guest interactions in the gas and condensed phases”
which reflexes the scope of the review more precisely. I suppose this title change will also make the review more appealing (and better cited) to a broader public interested in supramolecular chemistry of CDs in general and not only in mass-spectrometric aspects of it.
I have looked through the updated reference list and spotted that ref 47 definitely does not belong to the part on transition metal cluster binding, but to the essential oils part and should be moved into there. The phrase “On the other hand, it is well understood about the interaction of the noncovalent complex of CD/inorganic guest molecules…” (line 56) is not quite correct: it reflects well the situation of binding of metal cations, but the process of metal or borate cluster binding is much more complex and also encompasses weak dispersion interactions between CD and metal clusters. Thus, this part of the text should be updated. I would also recommend citing of the following recent papers relevant to this section: Phys. Chem. Chem. Phys., 2020,22, 7193-7200 & J. Am. Soc. Mass Spectrom. 2019, 30, 10, 1934–1945.
Several other minor comments:
- “because the solvent shielding by water molecules are absent.”- why “are”?
- “to elucidate the mechanism governing CD recognition in solutions like other host-guest chemistry.” – what is meant by “like other host-guest chemistry”
- “4. Noncovalent CD complexes containing gaseous guest molecules” – the whole complexes are in the gas phase, thus, more precise would be “Noncovalent CD complexes in the gas phase”
- In general, one additional proof reading performed by a native speaker could help to improve the manuscript stylistically at several places.
Otherwise, after the authors will consider these suggestions, the manuscript will be ready for publication.
Author Response
Responses to the reviewers’ comments
Reviewer 3
[Q1] After the author’s updates the content review looks much more balanced. Still, the major unbalance remains between the title - abstract and the contents of the review. Actually, it was the main reason of my criticism of the first version of the review: after reading of the title and the abstract, I expected to see a review focused almost exclusively on the MS studies, but the MS-related part of it started only on the page 8, which was almost half of the text of the review.
In my opinion, there is nothing wrong about having a chapter regarding the applications and studies of complex binding in the liquid/solid phase, but it should be reflected in the title and abstract of the review. My suggestions would be to announce other aspects of the review, namely, comparison of the condensed phase structures with the results obtained from the MS investigations, in the review abstract; currently the abstract it briefly mentions the applications and then almost completely focused on the MS studies.
In addition, I would suggest changing the title from:
“Interactions of cyclodextrins with small organic molecules in the gas phase: Experiments and Theoretical Studies” to something like (just a suggestion):
“Noncovalent complexes of cyclodextrin with small organic molecules: applications and insights into host-guest interactions {based on the comparative analysis of binding} in the gas phase and condensed phase”
or
“Noncovalent complexes of cyclodextrin with small organic molecules: applications and comparative analysis of host-guest interactions in the gas and condensed phases”
which reflexes the scope of the review more precisely. I suppose this title change will also make the review more appealing (and better cited) to a broader public interested in supramolecular chemistry of CDs in general and not only in mass-spectrometric aspects of it.
[Response] Thank you very much for your kind suggestions. First of all, the title of this review was changed from “Interactions of cyclodextrins with small organic molecules in the gas phase: Experiments and Theoretical Studies” to “Noncovalent complexes of cyclodextrin with small organic molecules: applications and insights into host-guest interactions in the gas phase and condensed phase”
Second, as you pointed out, we added some sentences in the abstract to reflect the contents regarding the applications and studies of complex binding in the liquid/solid phases.
Abstract: Cyclodextrins (CDs) have drawn a lot of attention from the scientific communities as a model system for host–guest chemistry and also due to its variety of applications in the pharmaceutical, cosmetic, food, textile, separation science, and essential oil industries. The formation of the inclusion complexes enables these applications in the condensed phases, which have been confirmed by NMR spectroscopy, X-ray crystallography, and other methodologies. The advent of soft ionization techniques that can transfer the solution-phase noncovalent complexes to the gas phase has allowed for extensive examination of these complexes and provides valuable insight into the principles governing the formation of gaseous noncovalent complexes. As to the CDs’ host-guest chemistry in the gas phase, there has been a controversial issue whether noncovalent complexes are inclusion conformers reflecting the solution-phase structure of the complex or not. In this review, the basic principles governing CD’s host-guest complex formation will be described. Applications and structures of CDs in the condensed phases will also be presented. More importantly, the experimental and theoretical evidence supporting the two opposing views for the CD-guest structures in the gas phase will be intensively reviewed. These include data obtained via mass spectrometry, ion mobility measurements, infrared multiphoton dissociation (IRMPD) spectroscopy, and density functional theory (DFT) calculations.
[Q2] I have looked through the updated reference list and spotted that ref 45 definitely does not belong to the part on transition metal cluster binding, but to the essential oils part and should be moved into there.
[Response] Thank you very much for your correction. The ref. 47 (the original ref. 45) was replaced with the following paper.
- Abramov, P.A.; Ivanov, A.A.; Shestopalov, M.A.; Moussawi, M.A.; Cadot, E.; Floquet, S.; Haouas, M.; Sokolov, M.N. Supramolecular Adduct of γ-Cyclodextrin and [{Re6Q8}(H2O)6]2+ (Q=S, Se). Clust. Sci. 2018, 29, 9–13.
[Q3] The phrase “On the other hand, it is well understood about the interaction of the noncovalent complex of CD/inorganic guest molecules…” (line 56) is not quite correct: it reflects well the situation of binding of metal cations, but the process of metal or borate cluster binding is much more complex and also encompasses weak dispersion interactions between CD and metal clusters. Thus, this part of the text should be updated. I would also recommend citing of the following recent papers relevant to this section: Phys. Chem. Chem. Phys., 2020,22, 7193-7200 & J. Am. Soc. Mass Spectrom. 2019, 30, 10, 1934–1945.
[Response] Thank you very much for your valuable suggestion. We revised the sentences as follow. We also added the recommended paper (reference 45).
Line 55-60:
“These complexities can be also found in the noncovalent interaction of CD/inorganic guest molecules including metal cations [38,39], polyoxometalates [40–43], dodecaborates [44–46], and octahedral rhenium clusters [47,48]. The three point rules are not applicable to explain the complexation of these molecules, in which more than three interaction points could exists or even no hydrophobic interaction takes place. Other references suggest that complementarity of the cavity size is also an important factor for the complexation.”
- Su, P.; Smith, J.A.; Warneke, J.; Laskin, J. Gas-phase fragmentation of host-huest complexes of cyclodextrins and polyoxometalates. Am. Soc. Mass Spectrom. 2019, 30, 1934–1945.
- Li, Z.; Jiang, Y.; Yuan, Q.; Warneke, J.; Hu, Z.; Yang, Y.; Sun, H.; Sun, Z.; Wang, X. Bin Photoelectron spectroscopy and computational investigations of the electronic structures and noncovalent interactions of cyclodextrin-: Closo -dodecaborate anion complexes χ-CD·B12X122- (χ = α, β, γ; F). Chem. Chem. Phys. 2020, 22, 7193–7200.
[Q4] “because the solvent shielding by water molecules are absent.”- why “are”?
[Response] Thank you for pointing out a typo. The sentence is modified as follows.
“Molecules in the gas phase undergo predominantly polar interactions such as hydrogen bonding and electrostatic interactions [26], which is due to the absence of the solvent shielding by water molecules. On the other hand, the hydrophobic interactions are weakened in the gas phase, in which the absence of water removes the driving force for the gathering of nonpolar surfaces [27,28].”
[Q5] “to elucidate the mechanism governing CD recognition in solutions like other host-guest chemistry.” – what is meant by “like other host-guest chemistry”
[Response] To clarify the meaning, the sentence was revised as follows, and a ref. 30 was added.
Line 33-34:
“The so-called “three points rule” or “lock and key model” has been applied to elucidate the mechanism governing molecular/chiral recognition of CDs in the solution phase [30].”
- Ariga, K.; Kunitake, T. The Chemistry of Molecular Recognition — Host Molecules and Guest Molecules. In Supramolecular Chemistry — Fundamentals and Applications; Springer, Berlin, Heidelberg, 2006; https://doi.org/10.1007/3-540-26185-0_2
[Q6] “4. Noncovalent CD complexes containing gaseous guest molecules” – the whole complexes are in the gas phase, thus, more precise would be “Noncovalent CD complexes in the gas phase”
[Response] We agree with your suggestion. It is corrected.
[Q7] In general, one additional proof reading performed by a native speaker could help to improve the manuscript stylistically at several places.
[Response] Thank you for your kind advice. The revised manuscript is again proof-read by a native speaker.